# PRACTICAL EPISTEMIC UNCERTAINTY QUANTIFICATION FOR VIEW SYNTHESIS

## ABSTRACT

View synthesis using Neural Radiance Fields (NeRF) and Gaussian Splatting (GS) has demonstrated impressive fidelity in rendering real-world scenarios. However, practical methods for accurate and efficient epistemic Uncertainty Quantification (UQ) in view synthesis are lacking. Existing approaches for NeRF either introduce significant computational overhead (e.g., "10x increase in training time" or "10x repeated training") or are limited to specific uncertainty conditions or models. Notably, GS models lack any systematic approach for comprehensive epistemic UQ. This capability is crucial for improving the robustness and scalability of neural view synthesis, enabling active model updates, error estimation, and scalable ensemble modeling based on uncertainty. In this paper, we revisit NeRF and GS-based methods from a function approximation perspective, identifying key differences and connections in 3D representation learning. Building on these insights, we introduce PH-DROPOUT, the first real-time and accurate method for epistemic uncertainty estimation that operates directly on pre-trained NeRF and GS models. Extensive evaluations validate our theoretical findings and demonstrate the effectiveness of PH-DROPOUT.

## 1 INTRODUCTION

Emerging approaches in view synthesis, such as Neural Radiance Fields (NeRF) (Mildenhall et al., 2021) and Gaussian Splatting (GS) (Kerbl et al., 2023b), have demonstrated remarkable advancements in rendering quality and efficiency. These methods transcend synthetic datasets to embrace real-world, unconstrained scenarios, setting new standards for generating highly realistic 3D scenes that are nearly indistinguishable from reality. However, achieving high-quality results requires a large number of known views for training, ensuring that the model is exposed to multiple perspectives near any arbitrary target viewpoint. Previous works (Goli et al., 2024; Sünderhauf et al., 2023; Hu et al., 2024) have highlighted that training view synthesis models from a discrete set of multi-view images is fraught with uncertainty. Even under ideal experimental conditions, occlusions and missing views inherently limit the epistemic knowledge that the model can acquire about the scene.

Studying epistemic uncertainty in view synthesis is crucial for understanding the limitations of algorithms, identifying gaps in model performance, improving the reliability of predictions, and ensuring effective generalization to unseen data. This investigation is pivotal for advancing the robustness and accuracy of view synthesis methods, yet research in this area remains limited. Broadly, related work can be classified into two categories: (1) Direct estimation of overall epistemic uncertainty, as seen in methods such as S-NeRF (Shen et al., 2021), CF-NeRF (Shen et al., 2022), and NeRF OTG (Ren et al., 2024), which apply deep ensemble techniques to NeRF with significant computational overhead. (2) Investigation of specific factors contributing to epistemic uncertainty, such as Bayes Rays (Goli et al., 2024) and CG-SLAM (Hu et al., 2024), which focus primarily on spatial (depth) uncertainty caused by a lack of views. In fact, epistemic uncertainty can be caused by a range of factors beyond mere lack of training data, including inadequate feature representation and model misspecification. Traditional methods for estimating epistemic uncertainty, such as Monte-Carlo dropout (Gal & Ghahramani, 2016) and random initialization (Lee et al., 2015; Lakshminarayanan et al., 2017), prove impractical for view synthesis due to conflicts with the training paradigm or the prohibitively high computational demands, requiring hours even for simple scenarios.

Another limitation is efficiency. Ensemble-based approaches can require hours of additional training for a single bounded object. Furthermore, there is a noticeable gap between GS and NeRF models, as existing methods lack versatility and are typically applicable to only one of these frameworks. However, we observe that NeRF-based and GS-based solutions excel in different scenarios. A versatile method integrating both approaches could enable the development of a new rendering technique that combines their respective strengths.

Driven by the goal of developing a more efficient, accurate, and versatile approach for estimating epistemic uncertainty in view synthesis, we revisit current view synthesis methods through the lens of function learning and approximation theory. We make the following key observations:

- Existing view synthesis models often exhibit **substantial parameter redundancy**; specifically, their performance on the training set remains unaffected by the application of appropriate dropout.
- However, while no observable impact is seen on training views, significant performance variance is evident on test views.

These insights lead us to propose a relatively underexplored post hoc epistemic uncertainty estimation method: (1) insert dropout on trained fully connected layers (NeRF) or splats (GS); (2) increase the dropout ratio as long as the performance on the training set remains unaffected; (3) measure the variation after dropout on the testing set as a quantification of epistemic uncertainty (UQ).

In summary, our contributions are:

- We propose PH-DROPOUT, the first approach, to our knowledge, that quantifies epistemic uncertainty for view synthesis in real-time, orders of magnitude faster, without additional training, and applicable to both NeRF and GS methods.
- We conduct comprehensive experiments with PH-DROPOUT on NeRF-based and GS-based methods, providing the first in-depth comparison of their features beyond training speed and fidelity.
- We evaluate the effectiveness of PH-DROPOUT through several downstream use cases, including active learning and model ensembling, where it demonstrates promising performance in supporting these applications.

## 2 BACKGROUND

### 2.1 EPISTEMIC UQ WITH SPECIALIZED TRAINING APPROACH

We explore general methods for epistemic UQ and explain why they are unsuitable for typical view synthesis tasks, highlighting the need for a novel approach. These traditional methods impose significant computational overhead and impose strict limitations on model selection.

*Random initialization* (Lee et al., 2015; Lakshminarayanan et al., 2017): is one technique used to measure epistemic uncertainty, particularly in the context of deep learning. However, it is not the most comprehensive method for capturing all aspects of uncertainty for following reason.

- **Training overhead**. Repeatedly retraining the model can be impractical, especially when the training process is slow or computationally expensive.
- **Random initialization cannot address limitations inherent in the model architecture itself**. If the model is not capable of learning the true underlying distribution, then random initialization won't reveal this inadequacy clearly.

*Monte Carlo Dropout* (Gal & Ghahramani, 2016): Applying dropout at test time and averaging predictions can be a cheaper way to estimate uncertainty without needing to train multiple models. NeRF's tendency to overfit arises from its high model capacity, the limited and specific nature of its training data, and its per-scene training approach. However, dropout is a method to prevent overfitting, which prevents NeRF to memorize the scene in training set. Empirically, we find that training with dropout in conventional NeRF places negative effect on performance, and hence MC dropout is not suitable for NeRF. Also GS models do not have a neural network, and hence hard to implement BNN based methods. Similar methods can be find in MC-batchnorm (Teye et al., 2018), where a deterministic network trained with batch normalization, which is also maintained during testing for UQ. Detailed discussion on applying MC-dropout on NeRF is enclosed in Appendix A.2

*Deep Ensemble Methods* (Lakshminarayanan et al., 2017): Training multiple models (not just with different initializations but potentially with different architectures or hyperparameters) and aggre-

gating their predictions can provide a more robust estimate of uncertainty. S-NeRF (Shen et al., 2021), CF-NeRF (Shen et al., 2022), and 3D Uncertainty Field (Shen et al., 2024) are based on this concept but with expensive training overhead and poor performance.

## 2.2 POST HOC EPISTEMIC UQ

Post hoc epistemic UQ refers to techniques for assessing a trained model's epistemic uncertainty without altering its original training process. The most frequently used frameworks for modeling uncertainty of neural networks are often not agnostic to the network architecture and task, and also require modifications in the optimization and training processes, including MC-dropout and deep ensembles. This is why post hoc UQ is desirable – it can be applied to already trained architectures.

We distinguish our study with the epistemic UQ in well pretrained models (Wang & Ji, 2024; Schweighofer et al., 2023). By well pretrained, the basic assumption of this method is the model is trained with sufficient in-distribution training data $\mathcal{D}$, so that $\forall x \in \mathcal{X} \rightarrow p(x \notin \mathcal{D}) < \epsilon$, where $\mathcal{X}$ is the set of potential inputs, $\epsilon$ is a small positive number. However, typical view synthesis tasks often face missing training views, making such methods unsuitable. Therefore, we exclude them as baselines in this paper.

Bayes Rays (Goli et al., 2024), based on Laplace approximation (Ritter et al., 2018), meets the post hoc requirements for practical application, requiring only a few additional training epochs. However, it solely models spatial uncertainty (depth prediction error) in NeRF and is not applicable to GS-based models. Previous work (Ledda et al., 2023) has also shown that in ad network based on fully connected layers, inject dropout during the inference time can achieve similar effect as MC-dropout (Gal & Ghahramani, 2016) after calibration. However, it is non-trivial to apply this method to view synthesis model. NeRF does not use dropout during the training, so it is hard to justify the approximation of MC-dropout. GS model even does not have a typical neural network.

## 3 PH-DROPOUT FOR EPISTEMIC UNCERTAINTY ESTIMATION

In this section, we first introduce the proposed algorithm in §3.1, then provide the conditions to ensure the effectiveness of the proposed method.

### 3.1 PH-DROPOUT

Here we introduce the proposed algorithm PH-DROPOUT. The process of PH-DROPOUT is illustrated in the following pseudo-code, where $F(x; \theta)$ is the trained rendering function (NeRF or GS)

---
**Algorithm 1** PH-DROPOUT

---
**Require:** Trained model: $F(\cdot; \theta)$, threshold $\epsilon$, step length $\Delta_r$, sampling number $N$
**Ensure:** $\mathbb{E}_x(|F(x; \theta) - F(x; \mathrm{PHD}(\theta, r))|) < \epsilon$
1: $r \leftarrow \Delta_r$          ▷ Initialize dropout ratio.
2: $D(\theta, r) \leftarrow M \cdot \theta$, $M_{ij} \in \{0, 1\}, \forall i, j$ and $r = \frac{\sum M_{ij}}{|\theta|}$ ▷ $M : \{M_{ij}\}$ is the binary dropout mask
3: $\mathbb{E}_x(|F(x; \theta) - F(x; \mathrm{PHD}(\theta, r))|) \leftarrow \frac{1}{N} \sum_i^N \sum_{x \in \mathcal{X}} \frac{|F(x;\theta) - F(x; D_i(\theta, r))|}{|\mathcal{X}|}$     ▷
     $D_i(\theta, r) \sim \mathrm{PHD}(\theta, r)$
4: **while** $\mathbb{E}_x(|F(x; \theta) - F(x; \mathrm{PHD}(\theta, r))|) < \epsilon$ **do**
5:      $r \leftarrow r + \Delta_r$          ▷ Increase dropout ratio
6: $r_{\mathrm{drop}} \leftarrow r - \Delta_r$          ▷ Select the maximal $r$ that meets requirement
7: **if** $r_{\mathrm{drop}} = 0$ **then**          ▷ $r_{\mathrm{drop}} = 0 \rightarrow$Wrong configuration
8:      Raise Error          ▷ The model is not properly trained
9: **else**
10:      $\zeta(x), \leftarrow \mathrm{std}(F(x; \mathrm{PHD}(\theta, r_{\mathrm{drop}})))$    ▷ Per pixel and channel UQ $\zeta$ of input $x$ based on std
11:      $\overline{\sigma_{\max}} \leftarrow G(\zeta(x))$          ▷ $G(\cdot)$ represents the processing in §A.1

---

with parameters $\theta$, $\mathbb{E}(\cdot)$ denotes the expectation, $\mathrm{PHD}(\cdot)$ refers to repeating stochastic inferences using independent dropout masks $M$, and $D(\cdot)$ is a sample of $\mathrm{PHD}(\cdot)$. It includes a heuristic solution of $\arg\max_r \mathbb{E}_x(|F(x; \theta) - F(x; \mathrm{PHD}(\theta, r))|) < \epsilon$. If dropout ratio $r_{\mathrm{drop}} = 0$ after interaction,

then this indicates the model is configured with less parameter than need (see Theorem 3.1). $r_{\text{drop}}$ is a measurement of parameter redundancy.

To further quantify the overall uncertainty $\zeta(x)$ of a model, we introduce $\overline{\sigma_{\max}}$ as a metric, which considers the max std across height, width, and RGB channels for each image, averaged over the whole rendered image set. See detailed definition in §A.1. By default, we use $\overline{\sigma_{\max}}$ to represent the overall uncertainty of a trained model on a given scene.

The differences to inject dropout in Ledda et al. (2023) are as follows,
- The dropout is applied with the condition that the model must still perfectly fit the training set afterward.
- Unlike a standard dropout layer, the dropout mask in PH-DROPOUT directly sets the weights (or splats in GS) to zero without scaling up amplitude of the rest weights.
- In NeRF-based methods, we apply dropout (i.e., a binary mask) to one of the middle layers, typically after the first fully connected layer, to selectively remove components from the render function, enhancing control over the process.

### 3.2 CONDITIONS OF USING PH-DROPOUT: FEATURES IN VIEW SYNTHESIS

The following phenomenons are integral to our reasoning and will be validated through experiments.

**Phenomenon 1**: The rendering function is not stochastic since we try to render a static object.

**Phenomenon 2**: After dropout, as long as there is no change on training set performance, the expectation performance on evaluation set only has negligible change as well (see §5).

Besides empirical observations, we also notice a common features across all NeRF-based and GS-based method: there must be redundancy in model parameters $\theta$, which is explained in detail in Theorem 3.1.

**Theorem 3.1.** *As long as the model is properly trained with overfitting ($\mathcal{L}(x) \to 0$ on training set), there must be significant redundancy in NeRF and GS model,* i.e.,

$$\exists\, 0 \ll r < 1 \to \forall x \in \mathcal{D}_{train},\ |F(x;\theta) - F(x;D(\theta,r))| < \epsilon$$

*Proof.* (**Sketch**) The rendering function is neither purely continuous nor purely discrete, but rather a combination of both. Achieving fine convergence using exclusively continuous methods (e.g., NeRF) or discrete methods (e.g., 3DGS) requires an infinite number of Fourier components (as in NeRF with positional encoding) or splats (as in splatting-based methods like 3DGS) to approximate functions with varying degrees of continuity.

*NeRF: Continuous function → discrete representation.* For many natural signals (Oppenheim et al., 1997), the amplitude of Fourier coefficients $c_n$ decay rapidly as $|n|$ increases, especially when signal is "continuously differentiable". This means that the lower-frequency components (those with smaller $|n|$) contain most of the signal's power, while the higher-frequency components (larger $|n|$) contribute very little to the total power. To overfit a function with nearly discrete pattern, many low power and high frequency components are introduced.

*GS: Discrete splats → continuous representation.* Similar to the Fourier transform, where fine spatial details (higher frequency components) generally have lower power, the training of GS models follows a similar pattern. A few large splats are used to capture the broader, background features, while numerous smaller splats are introduced to capture finer details and subtle variations in the rendering function.

As a result, in both NeRF and GS, to overfit the details of rendering function, most of the components have very low power and therefore are robust to dropout. □

Theorem 3.1 also highlights that NeRF-based and GS-based methods exhibit varying levels of redundancy at different regions of an object or scene. Both experimental results and theoretical analysis later demonstrate that this variation in redundancy patterns contributes to distinct robustness and reliability when capturing specific features. By leveraging an ensemble approach that selects the components with the lowest epistemic uncertainty, we achieve a significant improvement in view synthesis fidelity.

Empirical results show that all well-converged view synthesis models exhibit 20–30% redundancy, varying across methods. Removing this redundancy causes significant convergence issues during training. Thus, we conclude that PH-DROPOUT **does not require an additional increase in parameters, as this redundancy is by default required for proper convergence.**

## 4 EFFECTIVENESS OF ESTIMATION WITH PH-DROPOUT

### 4.1 EFFECTIVENESS ANALYSIS OF PH-DROPOUT IN NeRF

Here we discuss NeRF based methods with encoding that can reflect the spacial proximity faithfully, including Positional Encoding (PE) (Tancik et al., 2020), Sinusoidal PE (SPE) (Sun et al., 2024), etc. We discuss hash encoding-based methods separately in Section §4.3 due to the impact of their unique probabilistic operations.

**Lemma 4.1.** *If two models $a$ and $b$ with same structural and number of parameter, have similar distribution of parameter, **i.e.,** $D_{\mathbf{KL}}(\theta_a, \theta_b) < \epsilon$, and they both converge on the same dataset $\mathcal{D}$, they can be obtained via random initialization with the same setup with $a$ or $b$. Meanwhile, the probability density to obtain model $a$ and $b$ will be close,* i.e.*, if $p(a)$ is significant, then $p(b)$ should be significant as well.*

*Proof.* (**Sketch**). This can be proven based on the continuousity of the space of model parameter. The random initialization of weights of MLP (*e.g.,* in NeRF) from continuous distributions, which influences the training dynamics Glorot & Bengio (2010). These continuous distributions provide the network's starting point for training, and as training proceeds, the weights and biases are updated continuously by optimization algorithms (*e.g.*, stochastic gradient descent). These updates are applied to real-valued weights, thus ensuring that the neural network's parameters remain in the continuous space $\mathbb{R}^d$, where $d$ is the number of parameters Raghu et al. (2017). Because the function space $\mathbb{R}^d$ is continuous, if the two models have small KL partition, *i.e.*, $D_{\mathbf{KL}}(\theta_a, \theta_b) < \epsilon$, then $p(a) \approx p(b)$ in random initialisation because of the continuousity.                    □

With Lemma 4.1, we establish a connection between random initialization (Lakshminarayanan et al., 2017; Lee et al., 2015) and PH-DROPOUT, where each ensemble in PH-DROPOUT should exhibit significant probability density, assuming the trained model (w/o dropout) maintains substantial probability density within the function space. This forms the following Theorem.

**Theorem 4.2.** *The variance after* PH-DROPOUT *represents a biased epistemic uncertainty estimation.*

*Proof.* (**Sketch**) With PH-DROPOUT, the KL divergence of the remaining parameters from the original model is small. For instance, in NeRF, dropout is applied to only one hidden layer, leaving most parameters unchanged. Similarly, in the GS model, most splats have low power (see Proof of Theorem 3.1), and setting them to zero minimally impacts the overall power distribution.

We can further assume that the trained model is not an outlier and that the probability density of obtaining $F$ after random initialization is significant. Consequently, models after dropout represent a biased subset of the ensemble, and their variation captures epistemic uncertainty (see detailed discussion in Appendix A.3, Theorem A.2), as the probability density of these functions is non-negligible, consistent with Lemma 4.1.                    □

We have demonstrated that PH-DROPOUT is an effective approach for producing ensembles in NeRF, with the ensemble variation reflecting a biased estimation of epistemic uncertainty. Next, we extend this reasoning to the GS model.

### 4.2 EPISTEMIC UNCERTAINTY ESTIMATION IN GAUSSIAN SPLATS

**Theorem 4.3.** *During the training of 3DGS and 2DGS models following the scheme in (Kerbl et al., 2023a), the state of the splats in later training phase can be taken as changing in a continuous space,* i.e.*, the probability distribution density function $\mathcal{P}$ of model parameters $\theta$ is continuous.*

*Proof.* According to (Kheradmand et al., 2024), we know the typical splats updating scheme in (Kerbl et al., 2023a) can be approximated by a Stochastic Gradient Langevin Dynamics (Brosse et al., 2018), with a noise term missing. Standard Gaussian Splatting optimization could be understood as having Gaussians that are sampled from a likelihood distribution that is tied to the rendering quality. Suppose $\mathcal{P}$ is the data-dependent probability density function of models, it will have a form $\mathcal{P} \propto \exp(-\mathcal{L})$, where $\mathcal{L}$ is the loss function during training. Because the loss function $\mathcal{L}$ is continuous, the density function $\mathcal{P}$ should be continuous as well. □

Because of the continuousity of GS updating dynamics in Theorem 4.3, we can extend the Lemma 4.1 to typical GS models. Similar to dropout in the fully connected layers, we directly dropout the Gaussians, as they are the "weights" to optimize during the training.

### 4.3 PH-DROPOUT IS UNABLE TO HANDLE INPUT HASH COLLISION

NeRF may use hash encoding (HE) to process the input $x$. When collision happens, we have $x_i \neq x_j$ and $h(x_i) \approx h(x_j)$, where $h(\cdot)$ is the hashing operation. In HE based methods (Müller et al., 2022; Tancik et al., 2023), especially in few view cases, the model may not able to learn how to correct the hash collision due to lack of training data. In typical rendering tasks, much of the space is empty, a phenomenon leveraged by HE-based methods for more efficient learning. However, without proper supervision, the model may fail to render unseen regions, as the HE transformation brings the input too close to known empty space.

**Theorem 4.4.** *In sparse scenario, where most of the space is empty,*

$$p(F(x; \theta) > 0) \ll p(F(x; \theta) = 0), \ x \in \mathcal{X} \tag{1}$$

*where the RGB (final rendering) epistemic uncertainty caused by hash collision in HE cannot be detected by any ensemble based method, including* PH-DROPOUT*, $\mathcal{X}$ is set of potential inputs. The ensemble based method refers to epistemic uncertainty estimation by the variance of ensembles $F(x; \theta + \delta)$, where $|F(x, \theta) - \mathbb{E}_x[F(x; \theta + \delta)]| < \epsilon$, $\delta$ is the random perturbation.*

*Proof.* We extend the ensembles to hash encoding case as

$$|F(h(x); \theta) - \mathbb{E}_x[F(h(x); \theta + \delta)]| < \epsilon$$

Considering a significant portion of rendering output is empty, when hash collision happens on new input $x'$, we are very likely to have $F(h(x'), \theta) = 0$ according to Eq. 1. Because this is a rendering problem, the expected output must be positive, and so the ensembles $|\mathbb{E}_x[F(h(x'); \theta + \delta)]| < \epsilon$, which means the output of ensemble is almost zero. Therefore, as long as hash collision happens and the scenario is sparse, the output would be likely to be zero without variance, i.e.,

$$\mathrm{Var}[F(h(x'); \theta + \delta)] = \frac{1}{N} \sum (F(h(x'); \theta + \delta) - \bar{F}(h(x'); \theta + \delta))^2 < \epsilon^2$$

Near zero variance means no uncertainty, hence the epistemic uncertainty will not be reflected. □

Similarly, we can extend this conclusion to random initialization scheme. As long as the $h(\cdot)$ is a consistent pseudo hash function, the collision will happen at the same input, and will not give any informative information on the input with collision.

*Takeaway*. Due to hash collisions, faithful epistemic uncertainty estimation (on RGB) is not feasible when hash collision happens. Epistemic uncertainty in depth prediction is influenced by hash collision as well. See extended discussion about depth prediction and overall uncertainty in §A.7.

## 5 PERFORMANCE EVALUATION

### 5.1 EVALUATION ON TASKS

In rendering tasks, ground truth for epistemic uncertainty is unattainable, so we validate the effectiveness of uncertainty estimation indirectly through diverse tasks.

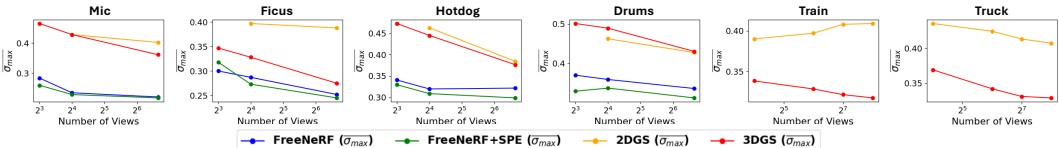

Figure 1: **Active Learning - $\overline{\sigma_{\max}}$**: PH-Dropout robustness to active learning is showed by a decreasing epistemic uncertainty at decreasing $\overline{\sigma_{\max}}$, with increasing number of training views.

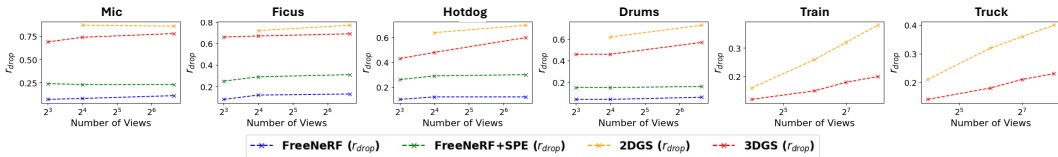

Figure 2: **Active Learning - $r_{\mathrm{drop}}$**: PH-Dropout robustness to active learning is showed by a decreasing epistemic uncertainty at increasing $r_{\mathrm{drop}}$, with increasing number of training views.

- **Active Learning** (Gal et al., 2017; Raj & Bach, 2022; Nguyen et al., 2022): Correlation with Training Data Sufficiency $\rho_{\mathrm{U}}$ and $\rho_{\mathrm{R}}$. A faithful estimation of epistemic uncertainty should show a lower uncertainty with a more training views. Therefore we take $\rho_{\mathrm{U}} = \rho_{\mathrm{s}}(\overline{\sigma_{\max}}, N_{\mathrm{train}})$ as metric, where $\rho_{\mathrm{s}}(\cdot)$ represents the Spearman's Correlation (Corder & Foreman, 2014), $N_{\mathrm{train}}$ is the number of training views, $\overline{\sigma_{\max}}$ is defined in Alg. 1. $\rho_{\mathrm{U}}$ **is expected to be negative**. Similarly, the redundancy (dropout ratio $r_{\mathrm{drop}}$ in Alg. 1) should increase as more training view is available, i.e., better overfitting according to Theorem 3.1. We introduce $\rho_{\mathrm{R}} = \rho_s(r_{\mathrm{drop}}, N_{\mathrm{train}})$ as the other metric reflecting correlation between model robustness and training views. Hence $\rho_{\mathbf{R}}$ **should be positive**. We streamline the estimation of the correlation coefficient by focusing on 8-view, 16-view, and 100-view setups, leveraging the overall trend for reasonable approximation.
- **Correlation with Prediction Error** (Liu et al., 2019; Nannapaneni & Mahadevan, 2016): $r_{\mathrm{PE}}$. Since the rendering function in view synthesis is deterministic, the primary sources of prediction error are epistemic uncertainty and model mis-specification. Therefore, a correlation between uncertainty estimation and actual error is expected. Specifically, we have $\rho_{\mathrm{PE}} = \rho_{\mathrm{s}}(\zeta(x), \mathrm{RMSE}(F(x), F_{\mathrm{GT}}(x)))$, where $\mathrm{RMSE}(\cdot)$ is the Root Mean Squared Error per pixel and channel, $F_{\mathrm{GT}}(x)$ denotes the ground truth image.

Besides effectiveness of PH-DROPOUT, we also highlight the efficiency of PH-DROPOUT in Figure 6. Even when focusing solely on inference speed, without accounting for other practical constraints in alternative methods, PH-DROPOUT demonstrates a performance gain of at least two orders of magnitude. This substantial efficiency improvement makes PH-DROPOUT the only viable option for use during runtime, with only a minimal frame rate drop on the initial render.

**Datasets**: We conducted experiments to evaluate the performance of PH-DROPOUT across 3 widely-used datasets: NeRF Synthetic Blender (Mildenhall et al., 2021), Tanks & Temples (T&T) (Knapitsch et al., 2017) and the LLFF dataset (Mildenhall et al., 2019).

### 5.2 ACTIVE LEARNING

For bounded case we evaluate the model on the Blender dataset following Mildenhall et al. (2021). The baselines for the bounded case of NeRF include FreeNeRF (Yang et al., 2023) and its fine-tuned variant FreeNeRF+SPE (Sun et al., 2024). These methods achieve superior fidelity in limited training view scenarios (few-view) and are free from hash collision issues, making them ideal benchmarks to demonstrate the effectiveness of PH-DROPOUT. GS based method includes 3DGS (Kerbl et al., 2023b) and 2DGS (Huang et al., 2024), yielding better fidelity and efficiency than NeRF based methods when there are sufficient training views. The results are demonstrated in Figure 1 and Figure 2 (detail in §A.8, Table 3), where the dropout ratio $r_{\mathrm{drop}}$ increases as the number of training view increases, indicating higher redundancy of the trained model. Meanwhile, the models tend to have higher uncertainty $\overline{\sigma_{\max}}$ when the number of training view decreases, even with smaller $r_{\mathrm{drop}}$. 2DGS is the only outlier on the trend of $\overline{\sigma_{\max}}$ because model collision in few view cases, §A.10.

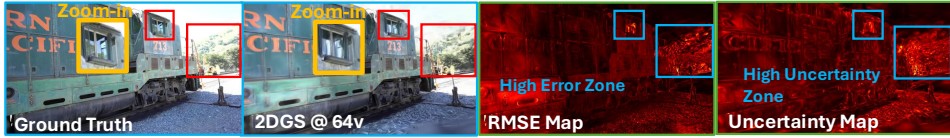

Figure 3: Correlation between PH-DROPOUT based epistemic uncertainty estimation and the actual RMSE, with 3DGS at bounded scenario (Blender drum). 8 training views.

Figure 4: Correlation between PH-DROPOUT based epistemic uncertainty estimation and the actual RMSE, with 2DGS at unbounded scenario. 64 training views.

FreeNeRF + SPE exhibits greater redundancy and lower UQ compared to FreeNeRF, despite both being based on NeRF with the same number of parameters. SPE (Sun et al., 2024) simplifies function learning by altering just one activation function (discussed in detail in the §A.4). This subtle structural change is clearly detected by PH-DROPOUT, further demonstrating the effectiveness of PH-DROPOUT.

For unbounded cases, we primarily focus on GS-based methods (2DGS and 3DGS), as conventional NeRF methods are too slow in this context without offering fidelity improvements. While HE-based NeRF is faster, it suffers from hash collisions in few-view setups, making it incompatible with other methods. Therefore, we explore NeRF for unbounded scenarios in a later section, where we implement PH-DROPOUT only on unbounded NeRF with sufficient training views. The results of GS-based models in unbounded scenarios, as shown in Figure 1 and 2 (detail in §A.8, Table 4), reveal a consistent pattern with the bounded scenarios. Specifically, as the number of training views increases, the dropout ratio $r_{drop}$ rises, while UQ metric $\overline{\sigma_{max}}$ decreases.

Combining results from both bounded and unbounded scenarios across NeRF and GS-based methods, we find a clear negative trend in $\rho_U$ and a significant positive trend in $\rho_R$. This demonstrates that the UQ provided by PH-DROPOUT is well-suited for supporting active learning tasks. This effective UQ is then applied to an uncertainty-driven ensemble usecase in §5.4.

### 5.3 CORRELATION BETWEEN UNCERTAINTY AND PREDICTION ERROR

As Figure 3 and Figure 4 illustrate, in 2DGS and 3DGS, high RMSE region tends to overlap with high uncertainty region. The correlation between RMSE and UQ in different scenarios is demonstrated in Figure 5. In bounded scenarios, both NeRF and GS-based methods demonstrate a strong correlation between RMSE and UQ. Results for 2DGS with 8 views are missing due to training limitations. In unbounded scenarios, GS-based models show lower correlation compared to bounded cases, reflecting the complexity of real-world data versus synthetic objects (Ren et al., 2024). Ad-

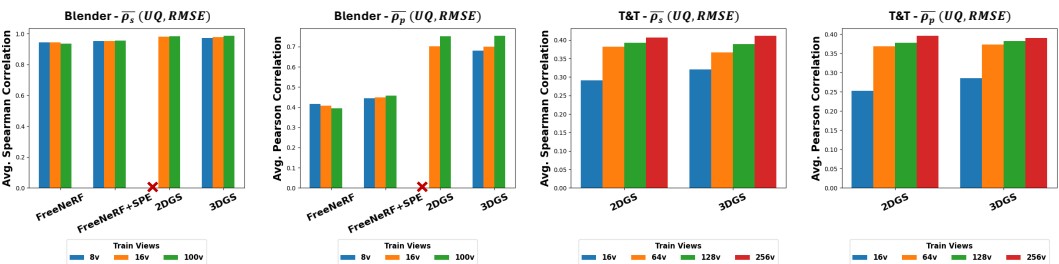

Figure 5: Correlation between RMSE and UQ of PH-DROPOUT in bounded (Blender) and unbounded (T&T) scenarios.

ditionally, both 2DGS and 3DGS exhibit higher correlation as training views increase, suggesting that insufficient training data leads to more unpredictable and random RMSE values. More detail about the correlation between PH-DROPOUT UQ and RMSE is showed in §A.9 Table 5 for bounded cases, and Table 6 for unbounded cases.

In addition to the methods discussed in Figure 5, we also explore hash encoding (HE) methods for error prediction, as they are not limited to few-view scenarios. For the unbounded case, we compare PH-DROPOUT against Bayes Rays (Goli et al., 2024) using its NeRFacto setup (Tancik et al., 2023), which, to our knowledge, represents the current state-of-the-art for unbounded NeRF and serves as an enhanced version of InstantNGP Müller et al. (2022).

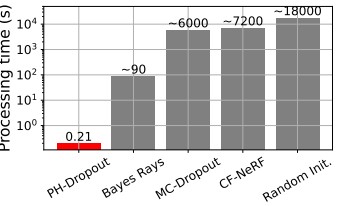

| Dataset | Method | PSNR ↑ | SSIM ↑ | $\rho_s$ ↑ | $\rho_p$ ↑ | AUSE RMSE ↓ | AUSE MSE ↓ | AUSE MAE ↓ |
|---------|--------|--------|--------|------|------|-----------|----------|----------|
| Africa | Bayes Rays | 22.1 | 0.839 | 0.020 | -0.071 | 0.545 | 0.512 | 0.512 |
| | PH-Dropout | 22.1 | 0.839 | **0.163** | **0.154** | **0.489** | **0.485** | **0.441** |
| Basket | Bayes Rays | 22.8 | 0.823 | -0.335 | -0.241 | **0.410** | **0.304** | **0.287** |
| | PH-Dropout | **22.9** | **0.825** | **0.342** | **0.310** | 0.438 | 0.345 | 0.351 |
| Torch | Bayes Rays | 24.4 | 0.867 | -0.395 | -0.196 | 0.454 | 0.314 | 0.348 |
| | PH-Dropout | **24.5** | 0.867 | **0.472** | **0.314** | **0.428** | 0.367 | **0.277** |
| Statue | Bayes Rays | 19.9 | 0.813 | -0.469 | -0.285 | **0.369** | **0.187** | **0.216** |
| | PH-Dropout | **20.0** | 0.813 | **0.370** | **0.166** | 0.596 | 0.469 | 0.468 |
| Avg. | Bayes Rays | 22.30 | 0.836 | -0.295 | -0.198 | **0.445** | **0.329** | **0.341** |
| | PH-Dropout | **22.38** | 0.836 | **0.337** | **0.236** | 0.488 | 0.417 | 0.384 |

Table 1: **Comparison of Bayes Rays and PH-DROPOUT on NeRFacto**: Bayes Rays fails to correlate depth uncertainty with high prediction error on the LF dataset.

Figure 6: PH-DROPOUT can be applied on-the-fly to a trained method, yielding orders of magnitude efficiency gain.

As for the comparison with baseline, we mainly focus on Bayes Rays (Goli et al., 2024), which carries out depth uncertainty of NeRF. Related work as NeRF OTG (Ren et al., 2024) is not considered as a baseline as it is proposed for dynamic scenarios. CG-SLAM (Hu et al., 2024) has recently explored similar spatial uncertainty-aware methods for GS models. However, it cannot be included either since CG-SLAM is not open-source and lacks sufficient implementation details. Computation-heavy methods like CF-NeRF (Shen et al., 2022) are excluded as they cannot be applied to more representative models, limiting the usefulness of their UQ. In Table 1, PH-DROPOUT achieves higher correlation ($\rho_s$ and $\rho_p$) between UQ and RMSE. This comes at no fidelity cost, empirically validating the negligible impact of inference-only dropout due to expected high model redundancy (see Theorem 3.1).

### 5.4 USECASES: UNCERTAINTY DRIVEN MODEL ENSEMBLES

Here we consider uncertainty driven model ensembling (Wang & Ji, 2023) as the usecase to further demonstrate the effectiveness of PH-DROPOUT. The ensemble method is driven by selecting the image with lower overall uncertainty, *i.e.*, select function $F$ from $F_a$ and $F_b$, following $\arg\min_{F \in \{F_a, F_b\}} \overline{\zeta_F(x)}$, where $\zeta_F(\cdot)$ is the per pixel and channel UQ under function $F$, $\overline{\zeta_F(x)}$ is the mean over all pixels and channel.

We randomly selected two non-overlapping 16-view training sets to train models '16v-a' ($F_a$) and '16v-b' ($F_b$). This task is challenging due to the random selection and the proximity between views in both sets, requiring the model to be highly sensitive in choosing the correct rendering results. As an ensemble method, we expect PH-DROPOUT to select the optimal view, ensuring overall fidelity that matches or exceeds the best ground-truth model between the two models.

Metric $E_{ME}$ is introduced to quantify the performance of ensembling. Here we consider model ensembling with dynamic selection, where two models with exactly same configuration are trained with different views of the same object, denoting as $F_a$ and $F_b$. We aim to evaluate the expected value of the following ratio, which directly reflects PH-DROPOUT's effectiveness in estimating uncertainty due to insufficient training views

$$E_{ME} = \mathbb{E}(r_{ME}) = \mathbb{E}_x \left( \frac{\text{SSIM}(F(x) | \arg\min_{F \in \{F_a, F_b\}} \overline{\zeta_F(x)})}{\max(\text{SSIM}(F_a(x), F_{GT}(x)), \text{SSIM}(F_b(x), F_{GT}(x)))} \right)$$

Intuitively, if the estimation can guide the selection of more suitable model, we will have $E_{ME} \to 1$.

| Method | | 2DGS | | | | 3DGS | | | |
|---|---|---|---|---|---|---|---|---|---|
| Dataset | Metric | 16v-a | 16v-b | selected | $E_{\text{ME}}$ | 16v-a | 16v-b | selected | $E_{\text{ME}}$ |
| Blender | SSIM | **0.887** | 0.873 | 0.881 | 0.974 | 0.900 | 0.894 | **0.912** | 0.996 |
| | PSNR | **24.4** | 24.1 | 24.3 | | | 25.4 | 25.3 | **26.3** | |
| T&T | SSIM | 0.564 | 0.553 | **0.591** | 0.952 | 0.550 | 0.537 | **0.601** | 0.987 |
| | PSNR | 15.4 | 15.6 | **16.3** | | 15.8 | 15.8 | **17.1** | |

Table 2: Performance on selecting rendered views with the lowest $\overline{\sigma_{\max}}$ from an ensemble of two models with different training views.

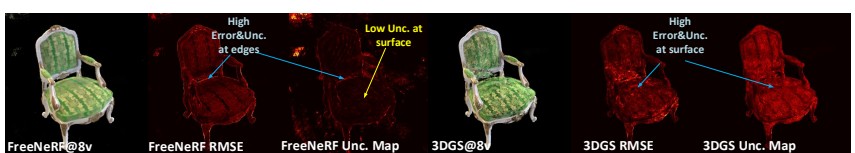

Figure 7: NeRF and GS models show distinct features in UQ and RMSE.

The detailed results are included in §A.11, Table 7 for bounded cases, where PH-DROPOUT successfully selects most correct views in 2DGS, with the ensemble model consistently performing at or near the level of the model with better training views. In 3DGS, PH-DROPOUT consistently selects the correct view, with $E_{\text{ME}}$ always close to 1. PH-DROPOUT performs better in 3DGS primarily because 2DGS experiences collision issues in few-view scenarios, leading to missing renderings, similar to hash collisions in HE-based methods. (see §A.10).

§A.11 Table 8 shows the ensemble performance in unbounded cases. PH-DROPOUT effectively selects images with superior fidelity, enabling the ensemble model to outperform any individual model. Similar to the bounded case, PH-DROPOUT shows reduced performance on 2DGS due to the inherent limitations of the 2DGS method.

# 6 DISCUSSION

## 6.1 DIFFERENT PERFORMANCE ON NERF AND GS, DIFFERENT ENCODING METHODS

Throughout this paper, NeRF and GS-based methods exhibit distinct patterns in redundancy and correlation with RMSE. As indicated by Theorem 3.1, these differences arise from the fundamental ways each method approximates the rendering function. Figure 7 illustrates that NeRF tends to show higher error and uncertainty at object edges, while GS models display increased uncertainty on smooth surfaces. This behavior aligns with the theorem and is a key factor behind the distinct UQ performance of each method.

## 6.2 LIMITATIONS OF PH-DROPOUT

PH-DROPOUT struggles with UQ in the presence of input hash collisions or similar model collisions in 2DGS, limiting its applicability to hash encoding-based methods (Müller et al., 2022; Tancik et al., 2023). Additionally, PH-DROPOUT is specifically designed for view synthesis tasks; further research is required to adapt it for other applications.

# 7 CONCLUSION

We present PH-DROPOUT, an efficient and effective epistemic uncertainty quantification (UQ) method for view synthesis, designed to operate directly on trained models. PH-DROPOUT is compatible with both NeRF and GS-based methods and can be applied to both bounded objects and unbounded scenarios. By offering fast inference and easy implementation, PH-DROPOUT makes epistemic UQ practical and stands as the first training-free method for UQ in GS models. Extensive evaluations across a broad range of downstream applications highlight its effectiveness. Theoretical analysis of PH-DROPOUT also uncovers fundamental differences and connections between NeRF and GS rendering methods, paving the way for future research to enhance their efficiency, fidelity, and scalability.

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

## A  APPENDIX

### A.1  COMPUTE THE STD OF A MULTI-CHANNEL IMAGE

Here we explain how to define the overall variance of the image $x$.

Let $\zeta(x) = \sigma \in \mathbb{R}^{N \times H \times W \times C}$ be a tensor representing the standard deviation image $x$ of $N$ views after $S$ stochastic forward-passes using PH-DROPOUT. The dimension of the rendered sampled images is $H \times W \times C$, where $H$ is the height, $W$ is the width, and $C$ is the number of channels.

For each view $i \in \{1, 2, \ldots, N\}$, we define the maximum standard deviation as:

$$\sigma_{\max,i} = \max_{h,w,c} (\sigma_{i,h,w,c})$$

where $h \in \{1, 2, \ldots, H\}$, $w \in \{1, 2, \ldots, W\}$, and $c \in \{1, 2, \ldots, C\}$.

We use the maximum value instead of the mean due to the sparse nature of the uncertainty map and the overall rendering process. A large portion of the pixels or space in the rendering is either empty or easily predictable, making the mean value ineffective for capturing meaningful variations, thus reducing sensitivity in quantification.

The mean of the maximum standard deviations across all $N$ views, denoted as $\overline{\sigma_{\max}}$, is defined as:

$$\overline{\sigma_{\max}} = \frac{1}{N} \sum_{i=1}^{N} \sigma_{\max,i} = \frac{1}{N} \sum_{i=1}^{N} \max_{h,w,c} (\sigma_{i,h,w,c})$$

## A.2 MC-DROPOUT IS NOT SUITABLE FOR NeRF

Overall, training dropout is a linear approximation of the average of ensembles, and hence prevent overfitting (Srivastava et al., 2014). The target rendering function is unique and deterministic, hence this technology cannot bring performance gain but only reduce the training efficiency and approximation accuracy.

**Theorem A.1.** *Training dropout prevents the efficient convergence of NeRF MLP on the rendering function.*

*Proof.* The color to render a ray $r$, *i.e.*, the rendering function is defined as:

$$C(r) = \int_{t_n}^{t_f} T(r)\sigma(r(t))c(r(t), d)dt$$

where $T(r) = \exp - \int_{t_n}^{t} \sigma(r(s))ds$ denotes the accumulated transmittance along the ray from from $t_n$ to $t$. Considering the typical NeRF with PE approximate rendering function using Fourier features should have a unique spectrum as: $C(r) = \sum w_i(r)\sin(r)$, $w_i(r)$ is the Fourier features. Applying dropout is equal to remove a few features, *i.e.*, $C'(r) = \rho \sum \mu_i w_i(r) \sin(r)$, where $\mu_i \sim \text{Bern}(p, N_f)$, $\rho = 1/(1-p)$. Since they both converge on training set, then on training set:

$$|\sum w_i(r)\sin(r) - \rho \sum \mu_i w_i(r)\sin(r)| < \epsilon \tag{2}$$

when $p = 0$ (no dropout), we have $\mu_i = 1$ as solution. However, when $p \neq 0$, an approximation can be made under certain conditions (ignore the empty space): (1) $p$ is small, so the dropout rate is low, ensuring that the majority of neurons remain active and the model behavior closely approximates the no-dropout scenario; (2) the power of each component is distributed sparsely, meaning the dropped components contribute minimally to the overall output, or the components exhibit even power distribution, akin to white noise.

Since the rendering function in most practical scenarios is not equivalent to white noise, it follows that the dropout ratio must remain small by default to avoid excessive loss of important information. Additionally, the distribution of power across components tends to be sparse in real-world cases, implying that only a few components carry significant influence.

This suggests that models with dropout can only effectively approximate cases where the components are sparse, leading to patterns that are simpler and lack fine-grained detail. As a result, while dropout helps prevent overfitting, it may also limit the model's capacity to capture intricate patterns when too much information is dropped. $\square$

Empirically, previous works Shen et al. (2024); Sünderhauf et al. (2023) have proven that the estimation of MC-dropout on NeRF is inaccurate with significant worse rendering quality when trained with dropout.

## A.3 CONDITION OF EFFECTIVE UNCERTAINTY ESTIMATION

**Theorem A.2.** *Suppose instances $F$ in a set of models $\hat{\mathcal{F}}$ (trained on same data $\mathcal{D}$), i.e.,*

$$\forall \hat{\mathcal{F}} \subset \mathcal{F}, |\hat{\mathcal{F}}| > T \to \frac{\sum_{\forall F \in \mathcal{F}} \hat{y}_F}{|\mathcal{F}|} = \frac{\sum_{\forall F \in \hat{\mathcal{F}}} \hat{y}_F}{|\hat{\mathcal{F}}|} + \epsilon_{mean}$$

*exhibit identical and perfect fitting performance on the training set, where $\hat{y}_F = F(x)$, $\epsilon_{mean}$ is the approximation error. The variation of their output can be interpreted as a reflection of epistemic uncertainty, if under a random initialization scheme $S(\cdot)$,*

$$\frac{|\hat{\mathcal{F}} \cap S(\mathcal{F})|}{|S(\mathcal{F})|} = \frac{|\hat{\mathcal{F}} \cap \mathcal{F}^*|}{|\mathcal{F}^*|} \gg 0$$

*where $\mathcal{F}^* = S(\mathcal{F})$ is the result of random initialization, assuming the data is non-noisy and deterministic.*

*Proof.* Let $\mathcal{F}$ be the set of all possible models that could explain the dataset $\mathcal{D}$. Each model $F \in \mathcal{F}$ provides a prediction $\hat{y}$ for given input $x$. We define function $F$ can explain a dataset if and only if

$$\forall (x, y) \in \mathcal{D} \to |y - F(x)| = |y - \hat{y}| \leq \epsilon$$

where $y$ is the expected and deterministic output to input $x$, appearing in pairwise in dataset $\mathcal{D}$

The predictive distribution of the model can be expressed as:

$$p(\hat{y}|x, \mathcal{D}) = \int_{\mathcal{F}} p(\hat{y}|x, F) p(F|\mathcal{D}) dF$$

The epistemic uncertainty can be then represented as

$$\mathrm{Var}[\hat{y}|x, \mathcal{D}] = \int_{\mathcal{F}} p(F|\mathcal{D})(\hat{y}_F - \mathbb{E}[\hat{y}|x, \mathcal{D}])^2 dF$$

where $\hat{y}_F \sim p(\hat{y}|x, F)$ and $\mathbb{E}[\hat{y}|x, \mathcal{D}]$ is the expectation of the prediction over the model posterior distribution.

To estimate the value of uncertainty $\mathrm{Var}[\cdot]$, we can conduct a Monte-Carlo solution. We first obtain an unbiased and significant number of instance $F$ forms set $\tilde{\mathcal{F}}$, and $\tilde{\mathcal{F}} \subset \mathcal{F}$. The expectation of prediction is estimated as $\frac{\sum_{\forall F \in \tilde{\mathcal{F}}} \hat{y}_F}{|\tilde{\mathcal{F}}|}$. The the estimated uncertainty is

$$\tilde{\mathrm{Var}}[\hat{y}|x, \mathcal{D}] = \sum_{F \in \tilde{\mathcal{F}}} \frac{N_F}{|\tilde{\mathcal{F}}|}(\hat{y}_F - \frac{\sum_{\forall F \in \tilde{\mathcal{F}}} \hat{y}_F}{|\tilde{\mathcal{F}}|})^2 = \mathrm{Var}[\hat{y}|x, \mathcal{D}] + \epsilon_{\mathrm{APP}}$$

$$\lim_{|\tilde{\mathcal{F}}| \to +\infty} \tilde{\mathrm{Var}}[\hat{y}|x, \mathcal{D}] = \mathrm{Var}[\hat{y}|x, \mathcal{D}]$$

where $\epsilon_{\mathrm{APP}}$ is the error caused by the bias of limited sampling number.

So far we have present the ideal case of uncertainty estimation. To obtain an ideal $\tilde{\mathcal{F}}$ is difficult because of the computation overhead and bias in sampling (*e.g.*, only consider $F$ with certain number of parameters). Here we discuss the feature of a subset $\hat{\mathcal{F}} \subset \tilde{\mathcal{F}}$.

Suppose the prediction expectation with $\hat{\mathcal{F}}$ has an error $\beta$

$$\frac{\sum_{\forall F \in \tilde{\mathcal{F}}} \hat{y}_F}{|\tilde{\mathcal{F}}|} = \frac{\sum_{\forall F \in \hat{\mathcal{F}}} \hat{y}_F}{|\hat{\mathcal{F}}|} + \beta(x)$$

the probability density of $F$ should be calibrate by $\alpha_F$

$$p(F|\mathcal{D}) = \frac{\alpha_F N_F}{|\hat{\mathcal{F}}|}$$

now the target estimation could be described with

$$\tilde{\mathrm{Var}}[\hat{y}|x, \mathcal{D}] = \sum_{F \in \hat{\mathcal{F}}} \frac{\alpha_F N_F}{|\hat{\mathcal{F}}|}(\hat{y}_F - \frac{\sum_{\forall F \in \hat{\mathcal{F}}} \hat{y}_F}{|\hat{\mathcal{F}}|} - \beta(x))^2 + \sum_{F \in \tilde{\mathcal{F}} - \hat{\mathcal{F}}} \frac{N_F}{|\tilde{\mathcal{F}}|}(\hat{y}_F - \frac{\sum_{\forall F \in \tilde{\mathcal{F}}} \hat{y}_F}{|\tilde{\mathcal{F}}|})^2$$

$$= \sum_{F \in \hat{\mathcal{F}}} \frac{\alpha_F N_F}{|\hat{\mathcal{F}}|}(\hat{y}_F - \frac{\sum_{\forall F \in \hat{\mathcal{F}}} \hat{y}_F}{|\hat{\mathcal{F}}|} - \beta(x))^2 + \delta(x)$$

$$= \sum_{F \in \hat{\mathcal{F}}} \frac{\alpha_F N_F}{|\hat{\mathcal{F}}|}(\hat{y}_F - \gamma(x) - \beta(x))^2 + \delta(x)$$

$$= \sum_{F \in \hat{\mathcal{F}}} \frac{\alpha_F N_F}{|\hat{\mathcal{F}}|} (\hat{y}_F - \gamma(x) - \epsilon)^2 + \delta(x)$$

where $\delta(\cdot) \geq 0$, $\beta(\cdot)$ is determined by $\hat{\mathcal{F}}$ and $x$, $\gamma(x) = \frac{\sum_{\forall F \in \hat{\mathcal{F}}} \hat{y}_F}{|\hat{\mathcal{F}}|} = \frac{\sum_{\forall F \in \hat{\mathcal{F}}} F(x)}{|\hat{\mathcal{F}}|} \geq 0$, and $0 < \alpha_F$.

The estimation on $\hat{\mathcal{F}}$ without calibration is

$$\hat{\text{Var}}[\hat{y}|x, \mathcal{D}] = \sum_{F \in \hat{\mathcal{F}}} \frac{N_F}{|\hat{\mathcal{F}}|} (\hat{y}_F - \gamma(x))^2$$

this is what we can compute directly.

Because all of the function in $\hat{\mathcal{F}}$ is equivalent to functions in $\mathcal{F}$, and $\tilde{\mathcal{F}}$ is also a subset of $\mathcal{F}$, then we have $\beta(x) \to \epsilon$ when $|\hat{\mathcal{F}}| \to +\infty$.

We can compute $\alpha_F$ as $|\hat{\mathcal{F}}|/|\tilde{\mathcal{F}}|$ if the size of both sets are available. The ratio of space $\alpha_F$ represents how the actual measurement contributes the ground truth uncertainty. And the actual uncertainty will be $\alpha_T V + \delta(x)$, as long as $\alpha_T \gg 0$, $V$ is an effective estimation, as large $V$ indicates high model uncertainty for sure.

Now we need to measure $|\hat{\mathcal{F}}|/|\tilde{\mathcal{F}}|$. Given the complexity of the space of the high dimension functions, we cannot easily compute the exact value. However, we can still verify the $\alpha_T$ is not a negligible small value by doing sparse sampling over $\tilde{\mathcal{F}}$ following reference schemes. Suppose there is a reference sampling scheme to obtain $\mathcal{F}^*$, if $1 > \frac{|\hat{\mathcal{F}} \cap \mathcal{F}^*|}{|\mathcal{F}^*|} \gg 0$, then the uncertainty measurement on $\hat{\mathcal{F}}$ represents the a significant part of uncertainty on $\mathcal{F}^*$. Also because of $1 > \frac{|\mathcal{F}^*|}{|\tilde{\mathcal{F}}|} \gg 0$, we have

$$\alpha_T = \frac{|\hat{\mathcal{F}}|}{|\tilde{\mathcal{F}}|} > \frac{|\hat{\mathcal{F}} \cap \mathcal{F}^*|}{|\mathcal{F}^*|} \cdot \frac{|\mathcal{F}^*|}{|\tilde{\mathcal{F}}|} \gg 0$$

By default, we obtain $\mathcal{F}^*$ via random initialization, which has been proven to be an effective way to represent the model uncertainty.

As a special case, if the data uncertainty at input $x$ is zero, then we must have

$$\delta(x) = 0, \forall F \in \hat{\mathcal{F}}, F(x) = \gamma(x)$$

This means, if the uncertainty is very low, then given arbitrary $\hat{\mathcal{F}}$ with a significant size $\frac{|\hat{\mathcal{F}}|}{|\tilde{\mathcal{F}}|} \gg 0$, we should have a stable $F(x)$.

If the model shows high uncertainty at input $x$ within set $\hat{\mathcal{F}}$, then this uncertainty will contribute to a significant part of the ground truth uncertainty. And if the model has overall low uncertainty at $x$, $F(x) - \gamma(x) \approx 0, \forall F \in \hat{\mathcal{F}}$ ◻

Random initialization is expensive and we cannot obtain $\mathcal{F}^*$ easily. Each trained model is just one instance of $\mathcal{F}^*$. Following aforementioned theorem, if we can find a subset of $\mathcal{F}^*$ with significant probability density, and guarantee the expectation of $\hat{y}$ converges to the global expectation with marginal error, then the estimation on this subset can reflect the lower bound of uncertainty.

## A.4 EXPLANATION OF DIFFERENCE BETWEEN SPE AND PE

Here we prove that conventional positional encoding (PE) (Tancik et al., 2020) needs more parameters to approximate the same function than sinusoidal positional encoding (SPE) (Sun et al., 2024). We first investigate the following question: how to use signal of frequency $f_1$ and $f_2$ to create a new frequency component. Suppose signal $y(t)$ is the weighted combination of sinusoidal signal with frequency $f_1$ and $f_2$, representing the initial fully connected layer with PE, $A$ is the amplitude

factor, and $F(\cdot)$ denotes the rest neural networks. To create a new frequency $\frac{f_1+f_2}{2}$, the following layers need performs a function $F^*(\cdot)$ to eliminate frequency $\frac{f_1-f_2}{2}$, as follows (because the signal is a combination of unique set of frequency features, $\frac{f_1-f_2}{2}$ may not be needed).

$$F(y(t)) = F(A\cos(2\pi\frac{f_1-f_2}{2}t)\sin(2\pi\frac{f_1+f_2}{2}t)) \tag{3}$$

$$= F^*(A\sin(2\pi\frac{f_1+f_2}{2}t)) \tag{4}$$

$$F^*(\cdot) = \cos(2\pi\frac{f_1-f_2}{2}t)^{-1}F(\cdot) \tag{5}$$

$$f = \sum_{i=0}^{L} w_i f_i, \text{where: } \sum w_i = 1, w_i \in \{m/2^n\}, m, n \in \mathbb{N} \tag{6}$$

Obviously, the highest freuqncy can be represented is bounded by

$$f \le f_L = 2^{L-1} \tag{7}$$

When $n$ is large enough, the error is bounded by $1/2^n$. Therefore, in theory, under ideal convergence, NeRF with PE can approximate arbitrary frequency within $2^{L-1}$ effectively.

However, when try to fine tune the high frequency features, the following effect will happen. The artifacts can be only be reduced when $f_1$ and $f_2$ are close. $f_l - f_{l-1} = f_{l-1}$ could be still require to learn high frequency representation directly via MLP. This results in NeRF is always struggling to approximate high frequency detail until the input sample rate is high enough (many views).

If $\frac{f_1+f_2}{2}$ is a new frequency features, then $\frac{|f_1-f_2|}{2}$ is a new frequency feature as well.

Following the standard PE, the input frequency component can be represented by $2^{L-1}$, the new created features is $(2^n+1)2^{m-1}$, so the synthetic frequency is always as Odd Num.$\times 2^n$ in pairwise.

$$\frac{2^m + 2^m \cdot 2^n}{2} = 2^{m-1} + 2^{m-1} \cdot 2^n \tag{8}$$

$$= (2^n + 1)2^{m-1} \tag{9}$$

in $\cos(\cdot)^{-1}$ side

$$\frac{2^m \cdot 2^n - 2^m}{2} = (2^n - 1)2^{m-1} \tag{10}$$

suppose now merge with $k$

$$\frac{(2^n+1)2^{m-1} + 2^k}{2} = (2^n + 1 + 2^{k-m+1}) \cdot 2^{m-2} \tag{11}$$

$$\frac{(2^n+1)2^{m-1} - 2^k}{2} = (2^n + 1 - 2^{k-m+1}) \cdot 2^{m-2} \tag{12}$$

Figure 8: NeRFacto (Tancik et al., 2023) fails to render the "dolls" due to hash collision. PH-DROPOUT reveals part of the missing dolls but cannot render the fully collapsed part. Depth Uncertainty wit Bayes Rays (Goli et al., 2024)

### A.5 OVER-CONFIDENT WITH PH-DROPOUT IN HASH ENCODING

This discussion tries to explain two phenomenon: (1) hash encoding based NeRF performs terrible in few-view cases, mixing object and background; (2) hash encoding is agnostic to PH-DROPOUT, *i.e.*, our method is less effective on hash encoding based NeRF.

In the hash encoding used in InstantNGP (Müller et al., 2022) and NeRFacto (Tancik et al., 2023), the input is encoded in a pseudo-random manner.

$$h(\mathbf{x}) = \left( \bigoplus_{i=1}^{d} x_i \pi_i \right) \mod T \tag{13}$$

where $\oplus$ denotes the bit-wise XOR operation and $\pi_i$ are unique, large prime numbers, $T$ is the size of hash table. Due to the pseudo-random encoding, similar encoded values do not necessarily reflect spatial proximity. Without targeted supervision, the MLP tends to regress based on the absolute values of the encoding, often producing similar results for close encodings. This occurs because the MLP, with ReLU (or other continuous activations), exhibits smoothness and continuity (Nair & Hinton, 2010; Yarotsky, 2018), making it difficult to effectively distinguish between background and object in few-view InstantNGP and NeRFacto.

*Takeaway*. In this paper, we apply PH-DROPOUT to hash encoding **only in many-view and unbounded scenarios** due to its overconfidence issue and poor generalization to unseen views.

### A.6 SUPPLEMENTAL RESULTS WITH HASH ENCODING BASED METHODS

We present a comprehensive comparison of three methods: NeRF + HE, NeRF + PE, and GS, using the Blender dataset as a benchmark. We evaluate their performance across two key scenarios: (1) rendering fidelity in a few-view setup (8 views and 16 views), and (2) training speed with a sufficient number of training views. These scenarios address two critical aspects of view synthesis approaches: how well the model generalizes when observations are limited, and how efficient the training and inference processes are when ample training data is available.

From the result in [], we observe a significant performance improvement in the few-view setup with the PE + NeRF method. This gain is primarily due to the method's ability to learn a continuous function, allowing it to capture low-frequency, large-scale features that generalize effectively across the spatial domain. In the second task, all methods show high fidelity rendering performance given sufficient training view. GS achieves highest training and inference efficiency.

For HE based methods, we notice a significant drop when implement them on PyTorch only. This is because HE based method uses indexing operations (*e.g.,* look up after hashing), which is easier to be accelerated with c++ compiler. As a result, a significant portion of efficiency gain of HE is caused by the difference between PyTorch and c++. And even under the ideal setup with c++, it is still slower than GS model.

*Takeaway*. HE-based methods receive less attention in this paper due to their poor performance in few-view setups and inefficiency in many-view scenarios.

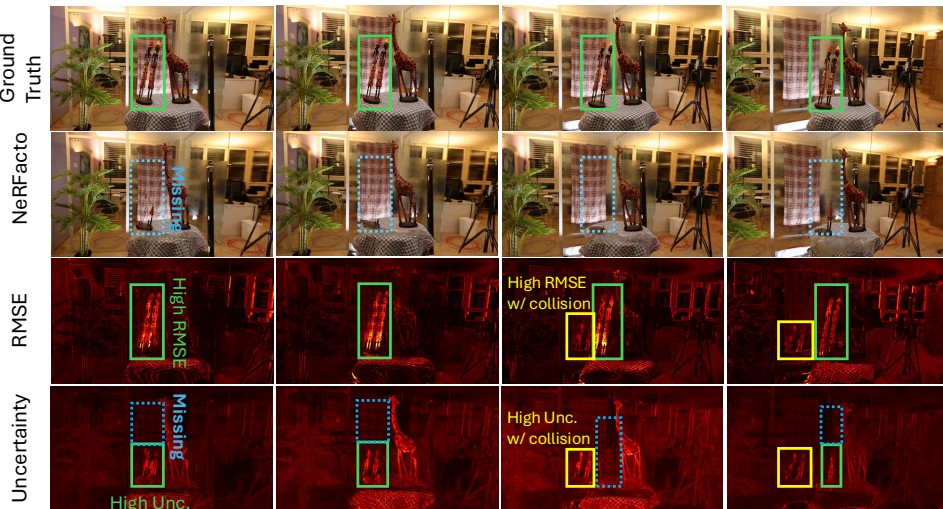

Figure 9: More views of PH-DROPOUT performance under hash collision. PH-DROPOUT can only track the rendered part, shift to left due the collision, highlighted with yellow

| Method | Metric | FreeNeRF 8v | FreeNeRF 16v | FreeNeRF 100v | FreeNeRF+SPE 8v | FreeNeRF+SPE 16v | FreeNeRF+SPE 100v | 2DGS 16v | 2DGS 100v | 3DGS 8v | 3DGS 16v | 3DGS 100v |
|---|---|---|---|---|---|---|---|---|---|---|---|---|
| | | Viewpoints (8v, 16v, 100v) | | | | | | | | | | |
| Mic | $\overline{\sigma}_{\max}$ | 0.283 | 0.234 | 0.220 | 0.259 | 0.228 | 0.217 | 0.428 | 0.402 | 0.465 | 0.428 | 0.361 |
| | $r_{drop}$ | 0.07 | 0.08 | 0.11 | 0.24 | 0.23 | 0.23 | 0.87 | 0.86 | 0.69 | 0.74 | 0.78 |
| Chair | $\overline{\sigma}_{\max}$ | 0.406 | 0.345 | 0.334 | 0.328 | 0.338 | 0.315 | 0.470 | 0.424 | 0.487 | 0.451 | 0.377 |
| | $r_{drop}$ | 0.08 | 0.07 | 0.10 | 0.19 | 0.19 | 0.20 | 0.60 | 0.60 | 0.47 | 0.50 | 0.55 |
| Ship | $\overline{\sigma}_{\max}$ | 0.339 | 0.284 | 0.288 | 0.306 | 0.313 | 0.298 | 0.427 | 0.417 | 0.430 | 0.382 | 0.346 |
| | $r_{drop}$ | 0.09 | 0.10 | 0.08 | 0.23 | 0.23 | 0.24 | 0.38 | 0.62 | 0.21 | 0.24 | 0.37 |
| Materials | $\overline{\sigma}_{\max}$ | 0.280 | 0.230 | 0.223 | 0.261 | 0.230 | 0.211 | 0.426 | 0.436 | 0.502 | 0.427 | 0.418 |
| | $r_{drop}$ | 0.08 | 0.10 | 0.13 | 0.26 | 0.28 | 0.30 | 0.65 | 0.72 | 0.49 | 0.52 | 0.57 |
| Lego | $\overline{\sigma}_{\max}$ | 0.344 | 0.354 | 0.325 | 0.363 | 0.352 | 0.348 | 0.461 | 0.382 | 0.463 | 0.417 | 0.339 |
| | $r_{drop}$ | 0.08 | 0.08 | 0.08 | 0.18 | 0.21 | 0.23 | 0.47 | 0.50 | 0.36 | 0.40 | 0.46 |
| Drums | $\overline{\sigma}_{\max}$ | 0.369 | 0.358 | 0.335 | 0.328 | 0.336 | 0.311 | 0.462 | 0.427 | 0.501 | 0.489 | 0.430 |
| | $r_{drop}$ | 0.04 | 0.04 | 0.06 | 0.15 | 0.15 | 0.16 | 0.62 | 0.73 | 0.46 | 0.46 | 0.57 |
| Ficus | $\overline{\sigma}_{\max}$ | 0.300 | 0.287 | 0.252 | 0.318 | 0.273 | 0.245 | 0.397 | 0.388 | 0.347 | 0.328 | 0.275 |
| | $r_{drop}$ | 0.08 | 0.12 | 0.13 | 0.25 | 0.29 | 0.31 | 0.72 | 0.77 | 0.66 | 0.67 | 0.69 |
| Hotdog | $\overline{\sigma}_{\max}$ | 0.340 | 0.319 | 0.321 | 0.329 | 0.308 | 0.298 | 0.463 | 0.384 | 0.473 | 0.445 | 0.376 |
| | $r_{drop}$ | 0.10 | 0.12 | 0.12 | 0.26 | 0.29 | 0.30 | 0.64 | 0.70 | 0.43 | 0.48 | 0.60 |

Table 3: **Active Learning Scenario on Blender dataset**: PH-Dropout robustness to active learning is showed by a decreasing epistemic uncertainty, $\overline{\sigma}_{\max}$, at a similar dropout rate $r_{drop}$, or a stable $\overline{\sigma}_{\max}$ at increasing $r_{drop}$, with increasing number of training views, given a constant $\epsilon$. The cases where PH-DROPOUT does not adhere to the active learning principle are marked with red.

## A.7 SUPPLEMENTARY EXPERIMENTS: UNABLE TO HANDLE HASH COLLISION IN HASH ENCODING BASED METHODS

In Figure 8 we show an example where NeRFacto does not render some objects, *e.g.*, the dolls highlighted with blue. PH-DROPOUT is able to show high uncertainty on the place NeRFacto tends to render but cannot show anything on the fully collapsed place. PH-DROPOUT still yields better robustness when collision happens, because the other methods require training as Bayes Rays will experience collision issue more significantly and fail to render anything on the collapsed regions. Figure 9 further demonstrates the influence of hash collision in HE. The yellow "ghost" effect is replication of the dolls, the NeRFacto model mix up two different input, and it does not consistent on the spatial domain because the collision is pseudo random.

## A.8 DETAILED RESULTS OF ACTIVE LEARNING TASK

Here we enclosed the detailed experiment results of the active learning usecase in §5.2, refer to the Table 3 and Table 4.

| Method | 2DGS | | | | 3DGS | | | |
|---|---|---|---|---|---|---|---|---|
| Metric | Viewpoints (16v, 64v, 128v, 256v) | | | | | | | |
| | 16v | 64v | 128v | 256v | 16v | 64v | 128v | 256v |
| Train $\overline{\sigma}_{max}$ | 0.390 | 0.397 | 0.408 | 0.409 | 0.338 | 0.328 | 0.321 | 0.317 |
| Train $r_{drop}$ | 0.16 | 0.26 | 0.32 | 0.38 | 0.12 | 0.15 | 0.18 | 0.20 |
| Truck $\overline{\sigma}_{max}$ | 0.435 | 0.424 | 0.413 | 0.407 | 0.369 | 0.342 | 0.331 | 0.329 |
| Truck $r_{drop}$ | 0.21 | 0.32 | 0.36 | 0.40 | 0.14 | 0.18 | 0.21 | 0.23 |

Table 4: **Unbounded Scenarios:** Performance of PH-DROPOUT on quantifying epistemic uncertainty in GS-based methods

| Method | FreeNeRF | | | | | | FreeNeRF+SPE | | | | | | 2DGS | | | | 3DGS | | | | | |
|---|---|---|---|---|---|---|---|---|---|---|---|---|---|---|---|---|---|---|---|---|---|---|
| Dataset | 8v | | 16v | | 100v | | 8v | | 16v | | 100v | | 16v | | 100v | | 8v | | 16v | | 100v | |
| | $\rho_s$ ↑ | $\rho_p$ ↑ | $\rho_s$ ↑ | $\rho_p$ ↑ | $\rho_s$ ↑ | $\rho_p$ ↑ | $\rho_s$ ↑ | $\rho_p$ ↑ | $\rho_s$ ↑ | $\rho_p$ ↑ | $\rho_s$ ↑ | $\rho_p$ ↑ | $\rho_s$ ↑ | $\rho_p$ ↑ | $\rho_s$ ↑ | $\rho_p$ ↑ | $\rho_s$ ↑ | $\rho_p$ ↑ | $\rho_s$ ↑ | $\rho_p$ ↑ | $\rho_s$ ↑ | $\rho_p$ ↑ |
| Mic | 0.981 | 0.412 | 0.985 | 0.396 | 0.984 | 0.397 | 0.984 | 0.435 | 0.983 | 0.425 | 0.982 | 0.415 | 0.996 | 0.693 | 0.996 | 0.750 | 0.996 | 0.720 | 0.997 | 0.754 | 0.997 | 0.776 |
| Chair | 0.963 | 0.388 | 0.962 | 0.338 | 0.967 | 0.407 | 0.967 | 0.405 | 0.967 | 0.400 | 0.970 | 0.412 | 0.990 | 0.739 | 0.993 | 0.817 | 0.989 | 0.682 | 0.991 | 0.742 | 0.994 | 0.830 |
| Ship | 0.873 | 0.454 | 0.863 | 0.438 | 0.856 | 0.414 | 0.887 | 0.484 | 0.879 | 0.458 | 0.891 | 0.486 | 0.940 | 0.694 | 0.943 | 0.683 | 0.898 | 0.643 | 0.915 | 0.619 | 0.943 | 0.642 |
| Materials | 0.918 | 0.329 | 0.934 | 0.361 | 0.868 | 0.164 | 0.939 | 0.372 | 0.943 | 0.400 | 0.945 | 0.410 | 0.979 | 0.680 | 0.982 | 0.695 | 0.964 | 0.669 | 0.982 | 0.669 | 0.984 | 0.678 |
| Lego | 0.936 | 0.413 | 0.933 | 0.406 | 0.927 | 0.419 | 0.944 | 0.434 | 0.946 | 0.455 | 0.949 | 0.461 | 0.979 | 0.696 | 0.986 | 0.769 | 0.975 | 0.671 | 0.979 | 0.692 | 0.987 | 0.792 |
| Drums | 0.961 | 0.472 | 0.947 | 0.378 | 0.953 | 0.392 | 0.963 | 0.497 | 0.959 | 0.453 | 0.961 | 0.474 | 0.980 | 0.649 | 0.990 | 0.703 | 0.980 | 0.660 | 0.988 | 0.652 | 0.991 | 0.673 |
| Ficus | 0.978 | 0.479 | 0.984 | 0.547 | 0.982 | 0.547 | 0.982 | 0.527 | 0.986 | 0.573 | 0.988 | 0.582 | 0.992 | 0.718 | 0.993 | 0.782 | 0.994 | 0.752 | 0.995 | 0.779 | 0.996 | 0.844 |
| Hotdog | 0.941 | 0.376 | 0.946 | 0.404 | 0.944 | 0.416 | 0.946 | 0.401 | 0.951 | 0.419 | 0.953 | 0.421 | 0.978 | 0.752 | 0.987 | 0.813 | 0.972 | 0.644 | 0.978 | 0.690 | 0.987 | 0.789 |

Table 5: **Bounded Scenarios**: Performance on Blender dataset with different training views. $\rho_s$: Spearman correlation, $\rho_p$: Pearson correlation.

## A.9 DETAILED CORRELATION BETWEEN RMSE MAP AND UQ WITH PH-DROPOUT

Here we include the per scenario correlation between RMSE and UQ in the following tables, Table 5 and Table 6.

## A.10 2DGS IN FEW-VIEW CASES

As Figure 10 shows, 2DGS encounters a similar issue to hash-encoding-based NeRF, where certain parts of the object fail to render entirely, limiting PH-DROPOUT's ability to detect significant variance. Without this variance, PH-DROPOUT cannot effectively perform uncertainty quantification (UQ).

## A.11 DETAILS OF ENSEMBLE USECASE

Here we enclose more details about the ensemble usecase in §5.4, including Table 7 and Table 8.

| Method | 2DGS | | | | | | | | 3DGS | | | | | | | |
|---|---|---|---|---|---|---|---|---|---|---|---|---|---|---|---|---|
| Dataset | 16v | | 64v | | 128v | | 256v | | 16v | | 64v | | 128v | | 256v | |
| | $\rho_s$ ↑ | $\rho_p$ ↑ | $\rho_s$ ↑ | $\rho_p$ ↑ | $\rho_s$ ↑ | $\rho_p$ ↑ | $\rho_s$ ↑ | $\rho_p$ ↑ | $\rho_s$ ↑ | $\rho_p$ ↑ | $\rho_s$ ↑ | $\rho_p$ ↑ | $\rho_s$ ↑ | $\rho_p$ ↑ | $\rho_s$ ↑ | $\rho_p$ ↑ |
| Train | 0.270 | 0.213 | 0.351 | 0.308 | 0.354 | 0.317 | 0.388 | 0.355 | 0.299 | 0.239 | 0.362 | 0.328 | 0.384 | 0.345 | 0.412 | 0.360 |
| Truck | 0.311 | 0.293 | 0.414 | 0.429 | 0.432 | 0.439 | 0.426 | 0.436 | 0.343 | 0.333 | 0.371 | 0.419 | 0.393 | 0.420 | 0.410 | 0.420 |

Table 6: **Unbounded Scenarios**: Performance of Gaussian Splatting methods on T&T dataset with different training views. $\rho_s$: Spearman correlation, $\rho_p$: Pearson correlation.

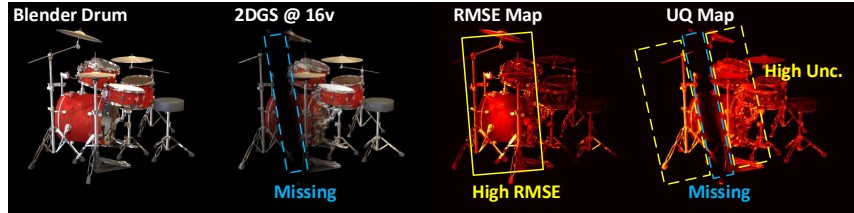

Figure 10: 2DGS misses rendering certain part of the object, when it is trained with few-view, *e.g.*, 16-views for blender dataset. Besides the missing part, PH-DROPOUT is able to show UQ with clear correlation with RMSE.

| Method | | | 2DGS | | | | 3DGS | | |
|---|---|---|---|---|---|---|---|---|---|
| Dataset | Metric | 16v-a | 16v-b | selected | $E_{\mathrm{ME}}$ | 16v-a | 16v-b | selected | $E_{\mathrm{ME}}$ |
| Mic | SSIM | 0.921 | 0.928 | **0.931** | 0.991 | **0.947** | 0.944 | **0.947** | 0.997 |
| | PSNR | 24.7 | **25.1** | 24.5 | | **27.4** | 26.9 | 27.2 | |
| Chair | SSIM | 0.920 | 0.927 | **0.935** | 0.991 | 0.931 | 0.935 | **0.938** | 0.992 |
| | PSNR | 25.6 | 26.0 | **26.1** | | 26.5 | 27.0 | **27.2** | |
| Ship | SSIM | **0.793** | 0.742 | 0.747 | 0.912 | 0.781 | 0.777 | **0.800** | 0.993 |
| | PSNR | **24.7** | 22.8 | 23.1 | | 25.5 | 25.2 | **26.1** | |
| Materials | SSIM | **0.871** | 0.857 | 0.863 | 0.978 | 0.892 | 0.884 | **0.903** | 1.00 |
| | PSNR | **23.3** | 21.7 | 22.6 | | 24.8 | 24.3 | **25.6** | |
| Lego | SSIM | 0.907 | 0.902 | **0.910** | 0.995 | 0.916 | 0.915 | **0.925** | 0.999 |
| | PSNR | 26.8 | 25.6 | **27.0** | | 28.0 | 27.6 | **28.3** | |
| Drums | SSIM | **0.859** | 0.777 | 0.818 | 0.935 | 0.890 | 0.851 | **0.901** | 0.998 |
| | PSNR | **19.9** | 18.9 | 19.8 | | 22.6 | 21.1 | **23.1** | |
| Ficus | SSIM | 0.917 | **0.933** | 0.924 | 0.988 | 0.932 | 0.935 | **0.936** | 0.998 |
| | PSNR | 23.7 | **25.7** | 24.5 | | 25.6 | 25.9 | **26.0** | |
| Hotdog | SSIM | 0.909 | 0.920 | **0.923** | 0.998 | 0.926 | 0.944 | **0.945** | 0.998 |
| | PSNR | 26.1 | 26.6 | **27.1** | | 26.9 | 29.5 | **29.7** | |
| Avg. | SSIM | **0.887** | 0.873 | 0.881 | 0.974 | 0.900 | 0.894 | **0.912** | 0.996 |
| | PSNR | **24.4** | 24.1 | 24.3 | | 25.4 | 25.3 | **26.3** | |

Table 7: **Bounded Scenarios.** Select synthetic view from models with different training views, so that two models are merged on-the-fly.

| Method | | | 2DGS | | | | 3DGS | | |
|---|---|---|---|---|---|---|---|---|---|
| Dataset | Metric | 16v-a | 16v-b | selected | $E_{\mathrm{ME}}$ | 16v-a | 16v-b | selected | $E_{\mathrm{ME}}$ |
| Train | SSIM | 0.476 | 0.491 | **0.522** | 0.940 | 0.463 | 0.468 | **0.526** | 0.979 |
| | PSNR | 13.3 | 14.5 | **14.9** | | 13.9 | 14.5 | **15.5** | |
| Truck | SSIM | 0.652 | 0.615 | **0.660** | 0.963 | 0.636 | 0.606 | **0.676** | 0.995 |
| | PSNR | 17.5 | 16.7 | **17.7** | | 17.7 | 17.0 | **18.6** | |
| Avg. | SSIM | 0.564 | 0.553 | **0.591** | 0.952 | 0.550 | 0.537 | **0.601** | 0.987 |
| | PSNR | 15.4 | 15.6 | **16.3** | | 15.8 | 15.8 | **17.1** | |

Table 8: **Unbounded Scenarios.** Select synthetic view from models with different training views, so that two models are merged on-the-fly.

