# OpenReview forum: "Practical Epistemic Uncertainty Quantification for View Synthesis"
_ICLR.cc/2025/Conference — Submitted to ICLR 2025_

### Official Review · Reviewer_xcSR · 2024-10-23

**Soundness:** 2
**Presentation:** 2
**Contribution:** 4
**Rating:** 6
**Confidence:** 4

**Summary:**

The authors propose using post-hoc dropout as a tool for epistemic uncertainty quantification in novel view synthesis frameworks like NeRF and Gaussian Splatting. Starting from a model with trained parameters, they uniformly increase the dropout rate of every parameter as long as the training loss is unharmed. Sampling from the distribution defined by this dropout rate can be used to compute spatially-varying variances in each novel view, which is used as a proxy for uncertainty, which is validated experimentally.

**Strengths:**

I congratulate the authors for their submission. The algorithm is simple, elegant and efficient. Previous attempts at using Dropout in novel view synthesis have failed: the authors have correctly identified the promise in the recent theoretically developed dropout injection methods and ported it to the Computer Vision community and the novel view synthesis problem in particular. Regardless of how exactly it compares against prior work, it surely advances the state of the art in uncertainty quantification for Vision and will inspire future work by the community (it has inspired *me* just by reading it!).

**Weaknesses:**

The main weakness of the current manuscript is the quality of the technical exposition. Given that what is proposed is a fairly simple extension of the Ledda et al. 2023 work, this poor quality of technical exposition is not enough to worry me about the soundness of the methods, but nonetheless the authors should strive to improve it in a revision, and this (together with the evaluation concerns below) is the reason why my score is not higher (I will gladly increase it if these questions are addressed). For example:

- The pseudocode in Page 3 has several mistakes. Line 4 should be before Line 1, and Lines 1-5 should be indented. The line starting with “Ensure” is redundant.
- The wording around \sigma_max  (L164-168) is confusing. See questions below
- The theorems and proofs in section 3.2 and 4 are incredibly vague, to the extent that I would be more comfortable if they were described as “intuitions” rather than “proofs” or “sketch of proofs”. A proof sketch is an abbreviated version of a proof for which one can be confident a motivated student reader can fill out the details. This is very much not that, as evidenced by my questions below. I would propose the authors rewrite these sections, abandon the pretence of mathematical correctness, explain their intuition and refer the reader to experiments for validation.
- After reading Lemma 4.1 and its proof 10 times, I still do not understand what it means. See questions below.

I would have also appreciated some visual examples of the uncertainty predicted in novel views by this method and its competitors that go beyond the numerical evaluations. By looking at numbers on a table, it is hard to understand what makes this method more precise than others. It claims to be “modeling all epistemic sources of uncertainty”, but this could be said of, e.g., the Goli et al. work as well, which is philosophically very similar (“how much can one modify network weights without harming the loss?”). In which image regions is this method doing a better work? Is this method better in floater removal? The text claims “Bayes Rays fails to correlate depth uncertainty with high prediction error on the LF dataset”, which seems to contradict the results in Goli et al.’s work. I would appreciate a deeper dive into this. This method is theoretically interesting and fast enough that I do not think its acceptance hinges on the evaluation being flattering, but I would expect a scientific paper to be more transparent in this regard and include a more exhaustive evaluation of where its results stand in relation to previous work, as well as its flaws and the flaws of previous works.

The manuscript would also benefit from some English proofreading (e.g., “continuosity” should be “continuity”), even if this has no bearing on my recommendation.

**Questions:**

I would love if the authors answered these questions in their revision:

- Why does the dropout mask in this method not need scaling the amplitude of the rest of the weights? I cannot think of a theoretical reason why one would do this, am I missing something? If not, is this due to an empirically observed advantage? I would be very curious to see this explored further, since it appears to go against the common wisdom of Dropout methods.
- About sigma_max (L164-168). Is the average being done over the training set only, or also the novel test view(s)? If the first one, isn’t the whole point of the algorithm that the training rendered RGB should not change after dropout injection, in which case sigma_max should be zero (or epsilon)? If the second one, how is something “quantifying the uncertainty of a model” if it depends on the specific view?
- What happens if the network is not perfectly overfit, as happens in almost every case? If the training is ended before the loss goes to zero, the network does not have sufficient resolution power, or it does but it is stuck in a local minima? The theoretical intuition seems to very deeply rely on this fact for the existence of redundancies: how much do the experiments suffer?
- The argument in Theorem 3.1 hinges on the conjecture that redundancy in the function space (e.g., one could modify the low power/high frequency Fourier components of this network without affecting the output much) translates into redundancy in the network parameters (i.e., one could turn network weights on and off without affecting the output much). I cannot immediately say that this conjecture is true (though it may very well be). Do the authors have any broader intuition or justification for this?
- On Lemma 4.1 and its proof: What is this D_KL in the space of weights? What does “p(a)” and “p(b)” mean? Are these conditioned on anything? I similarly do not understand the proof. Is all this lemma is doing saying that “if two weight sets are close, their Bayesian likelihoods are similar”? If so, this seems to me at least contestable. An argument about continuity may be made if the weights are changed only slightly, but if the observed redundancy is 20-30%, how can one state to be in a similar space in the Bayesian weight distribution?
- See “weakness” for questions on evaluation.

---

> ### Author Response · Authors · 2024-11-29
>
> Dear Reviewer pmy8
>
> Thank you again for your thorough reviews. Please find our responses to your questions and concerns below.
>
> ## To weaknesses
>
> > the pseudocode in Page 3 has several mistakes. Line 4 should be before Line 1, and Lines 1-5 should be indented. The line starting with “Ensure” is redundant.
>
> We will correct the improper sequencing of the various steps and address any writing issues to ensure clarity and accuracy.
>
> > "The theorems and proofs in section 3.2 and 4 are incredibly vague..."
>
> Thanks for your suggestions, and we will follow them to improve the writing of Section 4\. Sections 4 and parts of Section 3 will be reorganized around "observations" and "insights." Our approach relies on identifying suitable theorems or their combinations to reason about observed phenomena, rather than presenting new proofs.
>
> For example, in Line 190, we use the Fourier theorem to explain why sinusoidal components introduce redundant, low-power components when approximating a discrete function—this should be framed as intuition and analysis, not proof. Similarly, when explaining why dropout works in Gaussian splatting (Section 4.2), we rely on the observation that the number of splats is stable, allowing it to be interpreted as sampling from a continuous space—a concept quantified in prior work, not something to be proven. The misuse of "proof" has been noted by reviewers, and we will revise Section 4 accordingly to ensure proper formatting and clarity.
>
> > "I would have also appreciated some visual examples of the ..."
>
> We made substantial efforts to reproduce existing methods. Unfortunately, aside from Bayes-Ray, other baseline candidates like CF-NeRF and S-NeRF are either not open-source or lack visualization tools. We also contacted the authors of these two baselines through their GitHub repositories, but unfortunately, received no responses. Other teams attempting to reproduce these methods have encountered similar issues, as evidenced in the issues section on the CF-NeRF GitHub page. Despite these challenges, in the revision of our paper, we plan to include more visualizations of baselines in addition to Bayes-Ray and our method.
>
> > "It claims to be “modeling all epistemic sources of uncertainty”, but this could be said of, e.g., the Goli et al. work as well..."
>
> We will provide clarification and rephrase our claim to address this concern. Our aim is to model a broader range of epistemic uncertainty, which motivated the development of our method, particularly given that Bayes-Ray performs well in estimating depth uncertainty. Our approach extends beyond this, offering additional insights, such as explaining why 3DGS and NeRF produce distinct results using the same training data.

---

> ### Author Response · Authors · 2024-11-29
>
> ## To question (1)
>
> > Why does the dropout mask in this method not need scaling the amplitude of the rest of the weights?
>
> Scaling up after dropout during the training phase is crucial to maintain consistent activation levels and prevent learning bias. However, our method applies dropout solely during the test phase. In this case, the 1/p scaling factor, where p is the dropout rate, does not affect the relative uncertainty quantification (UQ) results across different inputs.
>
> While scaling up is necessary when focusing on absolute UQ values, empirical results from Ledda et al. (2023) show that post-hoc methods and MC-Dropout differ by a calibration factor C, for which no theoretical solution currently exists. Thus, in this paper, we do not aim to align absolute UQ values. Instead, relative UQ is sufficient to support most downstream applications.
>
> Skipping the scale-up step also slightly improves computational efficiency, though the gains are typically minimal. Our early empirical results align with Ledda et al. (2023), showing that relative metrics (e.g., correlation coefficients) remain unchanged with or without scaling.
>
> ## To question (2)
>
> > About sigma_max (L164-168).
> >“Is the average being done over the training set only, or also the novel test view(s)?”
>
> In fact, it does not fall under either of the two options.
>
> The sigma\_max is computed and evaluated on randomly selected test views, following the default settings of prior works like FreeNeRF. These test views are chosen with appropriate proximity to training views, effectively representing how trained models generalize to unseen views.
>
> >“which case sigma\_max should be zero (or epsilon)?”
>
> Exactly, it should be very small and close to zero, so we do not evaluate on training views, as they exhibit minimal variance.
>
> >“ if it depends on the specific view?”
>
> Uncertainty naturally depends on the test view; for example, views closer to training views typically show lower uncertainty. However, focusing on only a few anchor views would limit the generality of our results. Therefore, we evaluate trends across randomly selected test views from the literature while using specific viewpoints solely for visualization.
>
> ## To question (3)
>
> > “What happens if the network is not perfectly overfit, as happens in almost every case?”
>
> Our method may not work in this case. However, both empirically and theoretically, we clarify in the NeRF and Gaussian splatting based method there should be significant overfitting.
>
> > “If the training is ended before the loss goes to zero, the network does not have sufficient resolution power, or it does but it is stuck in a local minima?”
>
> We do not evaluate very early-stopping cases, as these are unrealistic in practical rendering tasks; our focus is on meaningful, well-trained models.
>
> However, we have considered the case of using training dropout to prevent overfitting. To the best of our knowledge, the model is properly trained, and dropout inherently mitigates overfitting. Further details are provided below (with the answer to the next question).
>
> > “...: how much do the experiments suffer?”
>
> Empirically, models that are not sufficiently “overfit” tend to exhibit overconfidence on arbitrary inputs. We train NeRF with dropout until it converges as much as possible, recognizing that dropout inherently prevents overfitting.
>
> The result is enclosed as following (on the blender dataset):
>
> Training Dropout rate: 0.2 (yields better performance), 50k interations, 16v for  training
>
> The implementation of FreeNeRF is based on DietNeRF, more details please refer to \[1\].
>
> | Method | Rendering SSIM | Rendering PSNR |$\rho_p$ |$\rho_s$ | $\sigma_{max}$|
> |------------------|-----------------|-----------------|-----------------|-----------------|-----------------|
> |FreeNeRF + PH-Dropout  | 0.879   | 24.341   | 0.42   | 0.945   |0.298 |
> |FreeNeRF + MC-Dropout  | 0.835   | 23.456   | 0.24   | 0.470   |0.091 |
>
> We observe that (i) NeRF trained with dropout shows worse fidelity — NeRF needs overfitting  (ii) the model trained with dropout is much less responsive to the testing phase dropout as the std is in general smaller. (iii) the std or variance shows low correlation to the actual error.
>
> Therefore, the conclusion is, our method requires the proper overfit to work properly.
>
> This experiment will be added to the final version of this paper.
>
> \[1\] J. Yang, “FreeNeRF: Improving Few-shot Neural Rendering with Free Frequency Regularization”

---

> ### Author Response · Authors · 2024-11-29
>
> ## To question (4)
>
> >"The argument in Theorem 3.1 hinges on the conjecture that..."
>
> Appendix A.4 provides material for a deeper understanding, explaining how sinusoidal components are combined by an MLP to approximate a high-dimensional function. Although its focus is on the difference between positional encoding (PE) and sinusoidal PE, the underlying mechanism remains the same.
>
> *Here’s a broader intuition:*
>
> Let’s start from a one layer case. In the simplest setup with a single-layer network, the fully connected layer's weights can be directly interpreted as the weights of the input Fourier components. With PE or SPE, the input x is expanded into a vector of sin(w⋅x), where w represents frequency components. Dropout on neurons in this scenario corresponds to removing specific Fourier features of x.
>
> In an MLP setup, each layer generates additional components (as discussed in A.4). However, the number of components effectively utilized cannot exceed the number of weights. Dropout applied to an intermediate layer introduces a bottleneck, potentially removing some Fourier components and limiting the function's expressiveness
>
> ## To question (5)
>
> >"On Lemma 4.1 and its proof: ..."
> “What is this D\_KL in the space of weights? ”
>
> By this, we aim to capture slight weight changes, which will be clarified in the revised version. We should avoid using D\_KL to describe differences between the weights of different models.
>
> >“if two weight sets are close, their Bayesian likelihoods are similar”? ”
>
>  Yes, the statement suggests that if two weight sets are very close and the model hyperspace is continuous, their Bayesian likelihoods should be similar.
>
> >“if the observed redundancy is 20-30%, how can one state to be in a similar space in the Bayesian weight distribution?”
>
> The relationship between redundancy and the proportion of total model parameters in PH-dropout differs between NeRF and Gaussian splatting methods. Below is a case-by-case clarification:
>
> For NeRF based methods, a redundancy level of 20%–30% does not imply that parameters are dropped randomly at this rate. Instead, dropout is applied to a single middle layer. In a typical NeRF setup with 8–10 layers, this affects only 2%–3% of the parameters overall, making it a ‘slight’ change. By targeting a single layer, we effectively remove Fourier features without significantly altering most model parameters. However, this explanation is based on intuition rather than formal proof, as noted. Section 4 will be revised for clarity.
>
> For Gaussian splatting methods, the continuity argument differs. Redundancy is lower at \~10% of all parameters, and prior work (Original 3DGS \[1\] and extension \[2\]) suggests that late training dynamics can be interpreted as MCMC sampling in the splatting space. Consistent dropout allows us to explore a continuous space of Gaussian splatting models with fewer parameters. According to \[2\], reducing parameters should not compromise fidelity. Again, this is an intuition rather than proof, which will be addressed in the revised version.
>
> \[1\] B Kerbl et al. ‘3D Gaussian Splatting for Real-Time Radiance Field Rendering’
>
> \[2\] S Kheradmand et al. ‘3D Gaussian Splatting as Markov Chain Monte Carlo’
>
> ## To question (6)
>
> > Revision plan for the evaluation part
>
> We will enclose more results on unbounded scenarios and quantitative results, as well as visualization of our method and baselines.
>
> We acknowledge the lack of source code for certain related works, such as CF-NeRF (as previously mentioned). Nonetheless, we will provide additional visualizations for reproducible methods like Bayes Ray and MC-dropout.

---

> > ### Comment · Reviewer_xcSR · 2024-11-30
> >
> > Thank you very much for your comments. I was positive about this work, and I remain positive. The proposed changes would indeed address most if not all of the issues I have raised.  The precise implementation of these changes feels to me important enough that I am maintaining my score as is, and I lament that the authors responded to reviewers only after the deadline to submit paper revisions had already passed.

---

> > > ### Author Response · Authors · 2024-11-30
> > >
> > > Thank you for your timely response.
> > >
> > > We will incorporate all the proposed revisions in the final camera-ready version. Due to time constraints during the rebuttal phase, we focused on addressing new experiments rather than fully polishing the paper.
> > >
> > > As the review process is public, we assure that all mentioned revisions will be fully incorporated into the final submission.

---

### Official Review · Reviewer_pmy8 · 2024-11-02

**Soundness:** 2
**Presentation:** 3
**Contribution:** 3
**Rating:** 6
**Confidence:** 3

**Summary:**

This paper proposes a Post Hoc Epistemic UQ scheme, referred as PH-DROPOUT, to make uncertainty estimation directly on pre-trained Multi view Scene Reconstruction models (NeRF and Gaussian Splatting in particular). The algorithm estimates the variance of a well-trained model by introducing binary dropout mask into model parameters and greedily select drop out ratio to perturb the output within tolerance, and validate over test views. Authors explore the usage of the measurement from their PH-DROPOUT algorithm by conducting thorough analysis over different scenarios (active learning, correlation analysis, uncertainty driven model ensembles).

**Strengths:**

Overall it is a good discovery and reflection on how to effectively and efficiently justify and analyze the performance of NeRF or GS model quantitatively. Authors justify their measurements is sound by extensive experiments and they indicate that most NeRF or GS models tend to have more redundant parameters as training views increase. This is a bold yet very inspiring claim. It implicitly conveys an intuitive thought: the information attained from more input views shall reduce the dimension of the model itself.

Mostly the paper context is well written and well organized, and the paper itself answers most of my concerns while I was reading it.

**Weaknesses:**

The math claim in this paper can be improved. There are too many "colloquial" proof rather than a mathematical modeling of phenomena. These claims seems to be redundant with respect to the integrity of the paper. For instance, to represent the significant redundancy in model parameters, authors state the following in line 190:

$$ \exists 0\ll r<1\rightarrow \forall x\in \mathcal{D}_{train}, \lvert F(x;\theta)- F(x;D(\theta,r))\rvert<\epsilon. $$

Firstly, greater than $\ll$ is not a rigorous notation. Secondly, $\mathcal{D}_{train}$ is not defined. Is it a point set or a multi-view image set? Lastly, the proof of line 190, i.e. line 192-210, is all text and there is no anchor point to refer to appendix for a comprehensive mathematical proof, and I failed to find the corresponding complete proof of theorem 3.1 in appendix. One cannot call statement in line 192-210 a proof of theorem. My suggestion is to make it as a conjecture or rewrite this theorem. Same question for lemma 4.1, thm 4.2, thm 4.3 and thm 4.4.

**Questions:**

There are some clarification questions I wish to hear from authors:

*    In line 141, the model function $F(x;\theta)$ is introduced. Is $x$ a 3D point? This confuses me when I checked line 190.
*    In Algorithm 1 (line 143-159), does $M\cdot \theta$ refer the element-wise product? Where does subscript $i,j$ come from?
*    In line 173-175, it claims dropout only happens in one of the middle layes in NeRF-based models? Any ablation study over this choice of selective layer dropout?
*    To show performance in active learning, the paper evaluates the performance starting from sparse view training (as low as 2 views). Nevertheless, PH-Dropout has an assumption that the pre-trained model needs to be well trained or over-fitted. I understand that some NeRF-based alrogithm can achieve sparse-view reconstruction. Is there any quantitative assessment on judging whether or not PH-Dropout is applicable in these sparse-view training results?
*    What do AUSE RMSE, AUSE MSE and AUSE MAE, mean in Table 1 (line 443-450). Besides, layout of Table 1 and Figure 6 is bad as title and floatings are not properly aligned.

---

> ### Author Response · Authors · 2024-11-29
>
> Dear Reviewer pmy8
>
> Thank you once again for your thorough reviews. Please find our responses to your questions and concerns below.
>
> ## To weaknesses
>
> >The usage of “\<\<”:
>
> We will remove ‘\<\<’. In fact, we initialized the value at 50%, based on empirical evidence that no model exhibits higher redundancy. The algorithm will then quickly optimize the dropout rate, leveraging the models’ fast inference speed.
>
> >“D\_train is not defined. Is it a point set or a multi-view image set?”
>
> It is a multi-view image set, just as the test set used in other 3D view synthesis works, e.g. 3DGS \[1\]. We will clarify this in the revised version.
>
> \[1\] B Kerbl et al. ‘3D Gaussian Splatting for Real-Time Radiance Field Rendering’
>
> >“the proof of line 190, i.e. line 192-210” and “thm 4.2 to 4.4”:
>
> Thanks a lot for the suggestion on section 4 revision. Overall, many of them are analysis based on intuition based on established theorems. And we would like to revise section 4\.
>
> In L190, we will change it to **observation** and **insight**, where the “**observation**” is the pixel wise representation of rendering is mixture of continuous and discrete pattern, and the “**insight**” is whether use continuous method (Fourier theorem and sinusoidal components in NeRF) or discrete method (point cloud or voxel), numerous components are required to approximate the rendering function. This leads to parameter redundancy, as most components contribute negligibly when rendering training views. This phenomenon suggests a method to generate multiple models that fit the training data, with their performance variance on test views reflecting epistemic uncertainty, as the definition of epistemic uncertainty. This conclusion relies on established theorems, such as the Fourier theorem, and does not require further proof. We will revise this section for clarity and smoother understanding.
>
> Similarly, in Section 4 (thm 4.2 to 4.4), much of the content is based on existing theorems and should be described as intuition rather than new proof. Additionally, some terms are used imprecisely. As noted by other reviewers, this section will be thoroughly revised to clarify where existing theorems are applied and to articulate the insights gained from combining them.

---

> ### Author Response · Authors · 2024-11-29
>
> ## To question (1)
>
> > * In line 141, the model function F(x;θ) is introduced. Is x a 3D point? This confuses me when I checked line 190\.
>
> It is a vector with at least five components, as defined in the typical rendering problem \[2\]. It includes 3D Coordinates (x,y,z) where the model needs to predict the radiance and density, and (the, phi), a unit vector or spherical coordinates representing the camera or ray's direction. We will add this clarification in the proper position.
>
> \[2\] B Mildenhall,  “Nerf: Representing scenes as neural radiance fields for view synthesis”.
>
> ## To question (2)
>
> > In Algorithm 1 (line 143-159), does M⋅θ refer the element-wise product? Where does subscript i,j come from?
>
> Yes, it is an element-wise product. In NeRF, *i* and *j* denote the layer number and elements within each layer. However, these indices are unsuitable for Gaussian splatting, which operates on a point cloud without "layers." To address this, we will remove the indices and clarify the dimensions of *M* in the text for each context.
>
> ## To question (3)
>
> > In line 173-175, it claims dropout only happens in one of the middle layers in NeRF-based models? Any ablation study over this choice of selective layer dropout?
>
> Yes, this is because it can produce the same result on NeRF based method with more robust and efficient performance (see details below). We will make this advantage clear in the revised version.
>
> First of all, applying dropout correctly across all layers does not improve correlation, as it primarily removes low-power components. NeRF approximates the rendering function using numerous Fourier features (see Appendix A.4 for details). The usable features are constrained by the hidden layer dimensions, with neural network weights corresponding to Fourier component weights. Consequently, applying dropout to a single layer can achieve the same effect as removing random Fourier components across all layers.
>
> Second, robustness is achieved by selectively dropping Fourier features while minimally altering parameters. For instance, in a 10-layer network, dropping 20% of weights in a single layer removes 20% of usable Fourier features. This however changes only 2% of the total weights. As suggested by Theorems 4.1 and 4.2, continuity is likely to be preserved under small parameter changes. In contrast, applying 2% dropout across all weights does not guarantee 20% feature removal and may significantly alter the combination of retained components. Therefore, single-layer dropout is preferable for robustness.
>
> Third, computation efficiency. Dropout on a specific layer provides efficiency gain, as we only need to compute the layers after the dropout.
>
> Corresponding ablation study is added. We will add more ablation study. Given the limited time in discussion phase, we take the bounded scenario NeRF and dropout cross all layers, the result is as follows:
>
> | Object     | $\rho_s$    | $\rho_p$    | $\sigma_{max}$  | $r_{drop}$ |
> |------------|-------|-------|-------|-------|
> | Chair      | 0.946 | 0.206 | 0.335 | 0.01  |
> | Drums      | 0.935 | 0.410 | 0.349 | 0.01  |
> | Ficus      | 0.987 | 0.610 | 0.245 | 0.06  |
> | Hotdog     | 0.958 | 0.442 | 0.291 | 0.05  |
> | Lego       | 0.954 | 0.472 | 0.332 | 0.04  |
> | Materials  | 0.946 | 0.446 | 0.220 | 0.06  |
> | Mic        | 0.982 | 0.405 | 0.213 | 0.06  |
> | Ship       | 0.847 | 0.404 | 0.271 | 0.03  |
> | Average (all layers)    | 0.944 | 0.423 | 0.282 | 0.04  |
> |------------|-------|-------|-------|-------|
> | PH-Dropout | 0.953 | 0.450 | 0.290 | 0.03 (0.24/8)  |
>
> The table shows that applying dropout across all layers reduces the correlation with actual error. Additionally, PH-Dropout produces higher variance with fewer dropped parameters, consistent with our analysis.
>
> ## To question (4)
>
> > "To show performance in active learning ..."
>
> The minimal number of training views required for reasonable rendering fidelity remains an open question. While our research is partially motivated by this issue, we cannot definitively answer it yet. Based on empirical observations, at least 8 views are needed for NeRF and 16 views for Gaussian splatting.
>
> ## To question (5)
>
> > What do AUSE RMSE, AUSE MSE and AUSE MAE, mean in Table 1 (line 443-450). Besides, layout of Table 1 and Figure 6 is bad as title and floatings are not properly aligned.
>
> AUSE refers to Area Under the Sparsification Error. A lower AUSE value indicates more reliable uncertainty estimates.
> Given an error metric (e.g. on in RMSE/MSE/MAE), we sort the prediction errors by their uncertainty in descending order and compute the error metric repeatedly by removing a fraction (e.g. 1%) of the most uncertain subset. An oracle sparsification curve is obtained by sorting using the true prediction errors. AUSE is the area between the sparsification curve and the oracle curve.
>
> We will clarify this in the revised version.

---

> ### Comment · Reviewer_pmy8 · 2024-12-02
>
> Thanks for your response. I will retain my score as well as my opinion that proofs and statements in theorems need revision. I would like to see these changes during discussion period. However, authors only offer proposed changes which I cannot tell whether it is good or bad.

---

> > ### Author Response · Authors · 2024-12-02
> >
> > Thanks again for your timely reply and your insightful reviews.
> >
> > Due to time constraints, we are unable to undertake a thorough revision on the manuscript with additional experiments at the discussion stage. However, we greatly appreciate your suggestions and will incorporate them comprehensively before the final version is released.

---

### Official Review · Reviewer_Ptfb · 2024-11-03

**Soundness:** 2
**Presentation:** 3
**Contribution:** 2
**Rating:** 6
**Confidence:** 4

**Summary:**

The paper proposes a simple yet effective and extremely fast ad-hoc method for epistemic uncertainty estimation that operates directly on pre-trained NeRF and GS models in the task of novel view synthesis. At its core, the method proposes to use dropout at test time with the maximal drop rate that maintains an epsilon fit to the training views. The method is validated through proxy metrics like Spearman’s correlation of the uncertainty estimates with the RMSE from the GT, and the trend as a function of the number of training views. In addition, the authors demonstrate improved ensembling based on optimal ensemble member selection using their uncertainty estimates.

**Strengths:**

* The authors considered multiple quantitative proxy metrics to benchmark their method, showcasing extensive evaluations.
* The method is very simple to implement and adopt, yet it is extremely effective and efficient in terms of runtime.
* Overall the paper is structured well and is easy to follow.
* Related work is acknowledged, and the contribution of this paper is put in proper context.

**Weaknesses:**

* Perhaps the major weakness of this paper is the relatively incremental contribution compared to Ledda et al 2023, which practically proposed the same technique up to the differences mentioned by the authors in L169-175. While I sincerely appreciate the honest citing of this related work by the authors, I find the contribution of PH-DROPOUT to be relatively small.
* The English writing of the paper can be significantly improved in certain spots (see small list below), although for the most part it is not hard to understand the authors intention based on the context.
* The proofs in the paper need to be more rigorous. For example, L192-207 are very hard to follow. Similarly, L235-243 seem to have mathematical inaccuracies such as calculating the KL divergence between two parameter instances instead of between two distributions. This overall proof sketch is not very clear and requires significant editing. The same applies to Theorem 4.2 and its proof sketch. More rigorous mathematical definitions and notations are needed. In the full proof from the appendix L777-782 seem to have a mistake in the variance formula. Where did $N\_F$ come from?

My rating is mainly due to the relatively limited contribution compared to Ledda et al. and the lack of mathematical rigor in the proofs. Nonetheless, I’m willing to increase my score if the authors' rebuttal can alleviate my concerns with respect to these two issues, as I still think this work does have the potential for practical value in quantifying the uncertainty of novel view synthesis.

A few caught (minor) typos/english corrections:

* L103 \- hence hard \-\> hence **it is** hard
* L104 \- find \-\> found
* L128 \- in **ad** network \-\> in **a** network
* L129 \- inject \-\> inject**ing**
* L130 \- **non-**trivial \-\> **not** trivial
* L149 \- step 3 in the algorithm is better split into 2 lines/formatted differently
* L162 \- need \-\> need**ed**
* L184 \- a common features \-\> common features/a common feature
* L185 \- method \-\> method**s**
* L198 \- signal \-\> **the** signal
* L201 \- what does a “function with nearly discrete pattern” mean?
* L208 of \-\> of **a**
* L225 \- spa**c**ial \-\> spa**t**ial
* L229 \- same structur**al** \-\> same **structure**
* L229 \- number of parameter \-\> number of parameter**s**
* L229 \- have similar distribution of parameter \-\> have **a** similar distribution of parameter**s**
* L235 \- continuou**sity** \-\> continuity
* ….

**Questions:**

* What is the number of dropout samples N in your experiments?
* Did you check the calibration of your uncertainty estimates (e.g. using metrics like ECE)?
* Did you check the trend of the RMSE from the GT compared to thresholding out pixels with increasing uncertainty levels? Is this expected to behave similarly to a NeRF model predicting both the mean and the standard deviation of the RGB value along each ray?
* In L262-264 you mention that your uncertainty estimate is biased. Is this bias not important? How does this affect your uncertainty calibration?
* When you write down $\\rho\_p$ in the results tables you are referring to $\\rho\_{PE}$?
* What does the acronym “AUSE” refer to in Table 1?

---

> ### Author Response · Authors · 2024-11-29
>
> Dear Reviewer Ptfb
>
> Thank you once again for your thorough reviews. Please find our responses to your questions and concerns below.
>
> ## To weakness (1)
>
> > While I sincerely appreciate the honest citing of this related work by the authors, I find the contribution of PH-DROPOUT to be relatively small.
>
> PH-Dropout employs a fundamentally different reasoning for using Dropout compared to Ledda et al. (2023). PH-Dropout is not an approximation of Monte Carlo (MC) Dropout, as proposed by Ledda et al. Instead, PH-Dropout creates ensembles by leveraging parameter redundancy, with an explanation of why such redundancy is prevalent in view synthesis. In contrast, Ledda et al. treat post-hoc Dropout as an approximation to MC-Dropout, which is inaccurate because training Dropout significantly influences training dynamics. This fundamentally different reasoning also leads to an oversight of a proper definition of model uncertainty in Ledda’s paper.
>
> These differences in reasoning have resulted in highly distinct algorithm implementations (L169-175). For example, PH-Dropout provides explicit methods for computing the Dropout rate and clear guidance on how to integrate Dropout into neural networks. In contrast, Ledda et al. do not provide algorithms or guidance on how to set the Dropout rate or implement Dropout in neural networks.
>
> ## To weakness (2)
>
> > The English writing of the paper can be significantly improved in certain spots (see small list below)
>
> We agree that some ‘proof’ in this paper should be ‘intuition’ built on existing theorems and new observations, and we will revise section 4 to reflect this.
>
> ## To weakness (3)
>
> **Clarification of L192-207:**
>
> L192-207 tries to show that numerous components are needed to approximate a rendering function with solely continuous representation (NeRF) and discrete representation (3DGS), because of the mixture of discrete and continuous features of the rendering function.
>
> For example, a discrete pattern corresponds to very high-frequency components in the Fourier transform, requiring NeRF to incorporate many Fourier components for approximation. Given a real-valued function with bounded output (e.g., 255 in RGB), the average power of components must decrease as their number increases. Thus, the observed redundancy in NeRF aligns with the Fourier theorem. This is an intuition analysis rather than not a proof.
>
> In summary, we agree that this is intuition on a developed theorem rather than a new theorem, and we will deeply revise this part.
>
> **Clarification of L235-243 and theorem 4.2:**
>
> In L235-243 and following theorem 4.2, the statement implies that if two weight sets are very close within a continuous model hyperspace, their Bayesian likelihoods should be similar. Thus, using D\_KL to describe differences between model weights is inappropriate. Instead, we aim to represent slight weight changes, which will be clarified in the revised version.
>
> **Clarification of L777-782:**
>
> In Appendix L777-782, N\_F is the number of function samples F during Monte-Carlo estimation. We will revise the text to introduce N\_F.
>
> Using Monte-Carlo method to estimate the epistemic uncertainty is used in MC-dropout \[1\] as well, and here the major difference is training dynamic to obtain the trained model and how dropouts are implemented.
>
> \[1\] Y Gal et al. ‘Dropout as a bayesian approximation: Representing model uncertainty in deep learning’
>
> **To the typos**
>
> And, thank you for very careful reviewing, we will fix all the typos in the revised version.

---

> ### Author Response · Authors · 2024-11-29
>
> ## To question (1)
>
> >What is the number of dropout samples N in your experiments?
>
> By default we use 30 samples. We also experimented with larger sample sizes, such as 100, and we observed no change in the conclusions. Therefore, we default to 30 samples for efficiency.
>
> ## To question (2)
>
> >Did you check the calibration of your uncertainty estimates (e.g. using metrics like ECE)?
>
> No, we haven’t. Specifically, ECE is for classification tasks, so we cannot apply ECE here.
>
> Here is the more fundamental reason for not doing this calibration check:
>
> Because our experiment is built on real rendering tasks in the wild, the ground truth of epistemic uncertainty is unavailable. Thus, it is hard to check the calibration in an absolute manner. (3) The metrics used in this paper are mainly some indirect measurement that shows whether the UQ is aligned with existing observation and supporting downstream applications.
>
> ## To question (3)
>
> >“Did you check the trend of the RMSE from the GT compared to thresholding out pixels with increasing uncertainty levels?”
>
> Yes, we did. The RMSE has positive correlation with the uncertainty level, it is shown in Figure 5, Page 8\.
>
> The correlation varies between dataset and models, up to \~0.93 Spearman correlation in bounded scenario NeRF and 0.35\~0.4 Spearman correlation in unbounded scenario. Unbounded real-world scenario shows lower correlation because the complexity in view specific features, such as highly reflective surfaces may only influence a few views that cannot be estimated from a training view anyway, recent real-world measurements \[2\] also point out the same challenge when correlated uncertainty with actual error.
>
> >“... behave similarly to a NeRF model predicting both the mean and the standard deviation of the RGB value along each ray?”
>
> No, it is not expected to perform like this. Even along the same ray, uncertainty can vary across spatial coordinates. This is because the coordinates are input into the probabilistic model (e.g., with PH-dropout), which is designed to capture the variance at each point rather than for the entire ray.
>
> \[2\] W Ren, et al. ‘NeRF On-the-go: Exploiting Uncertainty for Distractor-free NeRFs in the Wild’.
>
> ## To question (4)
>
> >Clarification: “uncertainty estimate is biased…”
>
> Bias cannot be properly quantified without a ground truth uncertainty measurement for reference. We know it exists and cannot quantify it using absolute values.
>
> Bias depends on the set of models being sampled. For example, MC-dropout samples only models trained with dropout, while our method samples models trained without dropout. Empirically, NeRF models trained with dropout exhibit significantly different dynamics, converge more slowly, and demonstrate slightly worse rendering fidelity (see attached results). Thus, MC-dropout and PH-dropout sample from distinct subsets of models with differing features, leading to inherent bias.
>
> >“Is this bias not important?”
>
> It is important to implement it. Quantifying the impact of bias requires ground truth uncertainty, which is unavailable.
>
> >“How does this affect your uncertainty calibration?”
>
> However, since representative methods like NeRF and Gaussian Splatting do not use dropout during training—and dropout does not improve performance (see table below)—we argue that our method is less affected by sampling bias. This extended discussion and results will be included in the revised version.
>
> Training Dropout rate: 0.2 (yields better performance), 50k interations, 16v for  training
>
> The implementation of FreeNeRF is based on DietNeRF, more details please refer to \[1\].
>
> | Method | Rendering SSIM | Rendering PSNR |$\rho_p$ |$\rho_s$ | $\sigma_{max}$|
> |------------------|-----------------|-----------------|-----------------|-----------------|-----------------|
> |FreeNeRF + PH-Dropout  | 0.879   | 24.341   | 0.42   | 0.945   |0.298 |
> |FreeNeRF + MC-Dropout  | 0.835   | 23.456   | 0.24   | 0.470   |0.091 |
>
> We observe: (i) NeRF trained with dropout shows worse fidelity — NeRF needs overfitting  (ii) the model trained with dropout is much less responsive to the testing phase dropout as the std is in general smaller. (iii) the std or variance shows low correlation to the actual error.
>
> Therefore, our method should be less affected by the sampling bias.

---

> ### Author Response · Authors · 2024-11-29
>
> ## To question (5)
>
> > When you write down p in the results tables you are referring to ρ_PE?
>
> It refers to Pearson correlation coefficient. We will make the definition more clear in the revised version.
>
> ## To question (6)
>
> > What does the acronym “AUSE” refer to in Table 1?
>
> AUSE refers to Area Under the Sparsification Error. A lower AUSE value indicates more reliable uncertainty estimates.
>
> Given an error metric (e.g. MAE), we sort the prediction errors by their uncertainty in descending order and compute the error metric repeatedly by removing a fraction (e.g. 1%) of the most uncertain subset. An oracle sparsification curve is obtained by sorting using the true prediction errors. AUSE is the area between the sparsification curve and the oracle curve.
>
> We will clarify this in the revised version.

---

> ### Author Response · Authors · 2024-12-02
> **Reminder**
>
> Dear Reviewer,
>
> I apologize for the delayed submission of my response. As today is the final day of the discussion period, I kindly ask if you could share any further comments on my rebuttal, particularly whether it resolves your concerns.
>
> Although I am unable to update the PDF at this stage, I assure you that all necessary revisions will be reflected in the final version of the paper.
>
> Thank you again for your insightful review and valuable feedback.
>
> Best regards

---

> > ### Comment · Reviewer_Ptfb · 2024-12-02
> > **Updated score**
> >
> > Thanks for the extensive answers. Like I said in my initial review, I like this method and think it is useful for the uncertainty community. Therefore, following your detailed answers and promised edits in the revised manuscript, I'm raising my score to 6 to enhance your chance of acceptance. Best of luck, and please incorporate all of the promised edits.

---

> > > ### Author Response · Authors · 2024-12-02
> > >
> > > Thank you for your timely response and insightful reviews.
> > >
> > > We are conducting a thorough revision of the paper, with a particular focus on Sections 3 and 4. We will ensure that all planned revisions are fully incorporated into the final version, enabling others to build on the phenomena and methods we have developed.
> > >
> > > Authors

---

### Official Review · Reviewer_6LNs · 2024-11-09

**Soundness:** 3
**Presentation:** 3
**Contribution:** 3
**Rating:** 6
**Confidence:** 2

**Summary:**

The paper introduces PH-DROPOUT, an efficient method for epistemic uncertainty quantification in view synthesis, applicable to pre-trained NeRF and GS models. It leverages model redundancy to estimate uncertainty without retraining, showing strong performance across datasets. Despite its efficiency, it faces challenges with hash encoding and sparse scenarios.

**Strengths:**

* The introduction of PH-DROPOUT as a post hoc epistemic uncertainty quantification (UQ) method is novel and addresses a critical gap in view synthesis research.
* The authors present a strong theoretical foundation, including proofs (e.g., Theorem 3.1) that justify the redundancy in NeRF and GS models, enabling effective dropout-based UQ.
* PH-DROPOUT achieves real-time performance, outperforming prior methods in computational efficiency.
* The method shows strong correlations between UQ metrics and prediction errors (e.g., RMSE), validating its reliability.

**Weaknesses:**

* The method struggles with hash encoding-based NeRF models, which restricts its applicability to sparse or complex scenes where hash collisions are common.
* The results on 2DGS show limitations in few-view scenarios.
* The method is specifically designed for view synthesis tasks and may not generalize to other domains.

**Questions:**

* Can the authors provide a workaround or mitigation strategy for handling hash collisions in NeRF models? This would address a key limitation of the method.
* The inclusion of more real-world datasets, particularly in unbounded scenarios, could strengthen the empirical evaluation.

---

> ### Author Response · Authors · 2024-11-29
>
> Dear Reviewer 6LNs
>
> Thank you once again for your detailed reviews. Below are our responses to your questions and concerns.
>
> ## To weakness (1)
>
> > The method struggles with hash encoding-based NeRF models
>
> This limitation arises from hash encoding (e.g., NGP-style NeRF), where the input neural network may experience random collapse. The only viable solution is to use sufficiently large hash tables.  However, this approach (large hash table) reduces the efficiency gains of NGP-style methods as the neural network's parameter count increases.
>
> Empirically, Gaussian splatting methods demonstrate better efficiency and fidelity, particularly with a large number of training views. Consequently, we will not focus on addressing the hash collision issue, as it is orthogonal to neural rendering technology and not a critical focus nor essential option.
>
> ## To weakness (2)
>
> > The results on 2DGS show limitations in few-view scenarios.
>
> This limitation affects uncertainty quantification and the rendering results, as the 2DGS model collapses in few-view scenarios. While model collapse (e.g., failure to render large regions, output zeros) is orthogonal to uncertainty quantification, we plan to investigate this phenomenon in future work and have highlighted it as a limitation in the newly added limitation section.
>
> ## To weakness (3)
>
> > The method is specifically designed for view synthesis tasks and may not generalize to other domains.
>
> There are domain specific analyses that may not be applicable to other domains. For example, the analysis on parameter redundancy in Section 3.2 can only apply to view synthesis.
>
> Besides the source of redundancy, there is no special assumption to be made by this method, hence we’d expect our method may work in other domains as well.

---

> ### Author Response · Authors · 2024-11-29
>
> ## To question (1)
>
> > “Can the authors provide a workaround or mitigation strategy for handling hash collisions in NeRF models?”
>
> The only solution is to use a sufficiently large hash table, which undermines the efficiency advantage of NGP-style methods.
>
> While this is an inherent limitation of the NGP approach, our method does not shed light on this issue. We highlight this as a limitation of NGP methods and clarify that our approach is orthogonal to input encoding, and therefore cannot resolve this problem.
>
> ## To question (2)
>
> We will add more results, this is also a concern mentioned by other reviewers. We are already working on more unbounded real world dataset during the discussion phase, here we enclose more results on unbounded scenarios, with Gaussian splatting etc.
>
> | Object    | Metric | 2DGS 16v | 2DGS 64v | 2DGS 128v | 3DGS 16v | 3DGS 64v | 3DGS 128v |
> |-----------|--------|----------|----------|-----------|----------|----------|-----------|
> | Bicycle   | $\sigma_{max}$   | 0.397    | 0.414    | 0.435     | 0.333    | 0.295    | 0.312     |
> |           | $r_{drop}$  | 0.14     | 0.28     | 0.27      | 0.10     | 0.13     | 0.11      |
> | Bonsai    | $\sigma_{max}$   | 0.471    | 0.398    | 0.357     | 0.425    | 0.376    | 0.290     |
> |           | $r_{drop}$  | 0.28     | 0.33     | 0.36      | 0.20     | 0.25     | 0.27      |
> | Counter   | $\sigma_{max}$   | 0.468    | 0.427    | 0.427     | 0.398    | 0.355    | 0.333     |
> |           | $r_{drop}$   | 0.31     | 0.42     | 0.47      | 0.19     | 0.20     | 0.24      |
> | Flowers   | $\sigma_{max}$   | 0.452    | 0.486    | 0.473     | 0.351    | 0.341    | 0.352     |
> |           | $r_{drop}$   | 0.19     | 0.44     | 0.26      | 0.09     | 0.12     | 0.09      |
> | Garden    | $\sigma_{max}$   | 0.410    | 0.395    | 0.387     | 0.344    | 0.312    | 0.338     |
> |           | $r_{drop}$   | 0.15     | 0.20     | 0.22      | 0.10     | 0.13     | 0.11      |
> | Kitchen   | $\sigma_{max}$   | 0.390    | 0.372    | 0.366     | 0.352    | 0.313    | 0.301     |
> |           | $r_{drop}$   | 0.20     | 0.26     | 0.26      | 0.17     | 0.22     | 0.22      |
> | Room      | $\sigma_{max}$   | 0.469    | 0.390    | 0.394     | 0.362    | 0.322    | 0.284     |
> |           | $r_{drop}$   | 0.37     | 0.41     | 0.47      | 0.22     | 0.27     | 0.29      |
> | Stump     | $\sigma_{max}$   | 0.415    | 0.426    | -         | 0.343    | 0.337    | -         |
> |           | $r_{drop}$   | 0.16     | 0.21     | -         | 0.11     | 0.12     | -         |
> | Treehill  | $\sigma_{max}$   | 0.462    | 0.484    | -         | 0.354    | 0.349    | -         |
> |           | $r_{drop}$   | 0.23     | 0.44     | -         | 0.10     | 0.12     | -         |

---

### Official Review · Reviewer_k2to · 2024-11-09

**Soundness:** 2
**Presentation:** 3
**Contribution:** 3
**Rating:** 6
**Confidence:** 4

**Summary:**

This paper proposes a fast post-hoc uncertainty quantification method for both NeRF and Gaussian Splatting Models. The method is based on test-time drop out of neurons/Gaussians in the model and estimating per-pixel uncertainty based on the error caused in the test views. The proposed method is orders of magnitude faster than other baselines and has competitive performance to them.

**Strengths:**

- Explores how drop-out as a main UQ approach in ML, fits into the radiance field framework.
- It is applicable to both NeRF-based and 3DGS-based models.
- The proposed method is very fast, and is done post-hoc which makes it a useful method for downstream applications.

**Weaknesses:**

Main:
- The rendering function in test time (the pre-trained radiance field) is not perfect. I do not see how comparing the rederings from the drop-out version of the model to the renderings of the model can specify uncertainty. Moreover, how are the test-views selected? Is there an assumption on the distribution of the cameras? If you select your camera far enough from the training ddistribution, the dropout rate would drop to zero , as the error from that view would be high anyway?
- The  volume rendering function itself (through its integration) hides the ambiguity and uncertainty in depth, so using the rendered image error as a source for identifying uncertainty can be less robust than a spatial method, unless queried densly from many test views.
- The authors claim this method is not architecture-dependant, a discussion about how drop-out in MLP-based NeRFs vs voxel-based or K-Plane ones results in the same metric would be useful. My confusion comes from the fact that some of these representations have more geometric meaning (like voxel or K-Planes or 3DGS) and in these cases dropping out a cell would directly affect a point in space while dropping out a node/layer from MLP might affect different places simultaneously.
- Qualitative results on uncertainty primarily reflect color uncertainty, as seen in flat regions like the train body in figure 4, where low-opacity, cloudy Gaussians exhibit low uncertainty. However, the paper should clarify whether the reported uncertainty correlates more with color error, depth error, or both, and include qualitative evidence to support this claim. This helps readers understand what use cases this uncertainty is useful for.
Minor:
- What are the PSNRs reported in Table 1? Are they PSNR after noise removal the same way explored in the baseline, if so the coverage % should also be reported alongside this metric.
- A few results on NGP-style NeRFs which are the main NeRF models used, and most results on MLP NeRFs.
- The proof sketches in the main paper can be more detailed.

**Questions:**

See above.

---

> ### Author Response · Authors · 2024-11-29
>
> Dear Reviewer k2to
>
> We would like to thank you again for your detailed review. Below, we answer your questions and address your concerns.
>
> ## To weakness (1)
>
> >“how are the test-views selected?”
>
> PH-Dropout does not select test-views. This is because, the camera/test-view selection, it works with arbitrary views, where we use the random selected test views as per base methods for the convenience of the result reproduction.
>
> >“Is there an assumption on the distribution of the cameras?”
>
> PH-Dropout does not need an assumption of the selection of test views.
>
> >“ If you select your camera far enough from the training ddistribution, the dropout rate would drop to zero , as the error from that view would be high anyway?”
>
> PH-Dropout does not encounter this issue because the dropout rate is independent of the camera’s position. The dropout rate is calculated using Algorithm 1\. Specifically, as outlined in lines 4–6 of Algorithm 1, the dropout rate is determined by applying dropout to the train views, using the highest rate that does not affect rendering performance on these views. Empirically, the dropout rate is not negligible, typically ranging from 10% to 30%, and varies depending on the base method.
>
> This dropout rate computation process is applied consistently to all tested views. Consequently, even for distant views, the dropout rate remains significant, albeit with much higher variance (in contrast, there should be near-zero variance for train views). This results in greater uncertainty for far-away test views.

---

> ### Author Response · Authors · 2024-11-29
>
> ## To weakness (2)
>
> In fact, PH-Dropout is more robust than a spatial method. The primary reason is “uncertainty in depth” is just one of the factors that influences the Model (Epistemic) Uncertainty.
>
> For instance, in addition to spatial blockage, model structure could also affect model uncertainty. To further explain this, we could consider the Spatial Uncertainty (e.g., Bayes Ray) which cannot explain the fact that 3DGS and NeRF give distinct rendering even trained on the same views. However, our method can show the uncertainty caused by the model structure in Figure 7, page 10\.
>
> Empirically, even on depth uncertainty estimation, overall our method achieves the same level of accuracy as Table 1 shows.
>
> > “unless queried densly from many test views”
>
> We focus on the rendered image error because it is the final output of the model.
>
> ## To weakness (3)
>
> > “The authors claim this method is not architecture-dependant”
>
> PH-Dropout does not claim to be “architecture-independent” or “model-agnostic,” as none of these terms or related keywords appear in the draft. Instead, we emphasize that PH-Dropout demonstrates stronger generalization capabilities than prior methods. This is evidenced by its competitive and often superior performance in both NeRF and Gauss Splatting scenarios compared to their respective state-of-the-art methods.
>
> > “ My confusion comes from the fact that some of these representations have more geometric meaning (like voxel or K-Planes or 3DGS)”
>
> We apologize for any confusion our draft may have caused and would like to clarify this in more detail:
>
> For voxel-based methods, InstantNGP or NeRFacto are the representative approaches, encoding the space as voxels. Our method achieves state-of-the-art (SOTA) performance in this category (Page 10, Table 1).
>
> K-plane-based methods differ from NeRFacto by encoding space into a few 2D planes instead of 3D voxels. Our approach can also be applied to the neural network component of K-plane, as demonstrated with NeRFacto. Due to time constraints, we could not provide quantitative results, and we plan to incorporate K-plane into the revised version.
>
> ## To weakness (4)
>
> Given the limitations of GPU resources available to us, we have made our best effort to include as many datasets as possible in our experiments. Nevertheless, the datasets we included are still comparable to those leading papers published in related fields, such as NeRF \[1\], FreeNeRF \[2\], and 3DGS \[3\]  cited in our paper.
>
> Since the paper submission, we have successfully run new datasets such as LLFF etc. We are currently in the process of open-sourcing these additional results, which is a non-trivial task due to the significant cloud storage space they require (e.g., results of variational inference).
>
> \[1\] B Mildenhall, “Nerf: Representing scenes as neural radiance fields for view synthesis”.
>
> \[2\] J Yan, “Freenerf: Improving few-shot neural rendering with free frequency regularization”.
>
> \[3\] B Kerbl et al. ‘3D Gaussian Splatting for Real-Time Radiance Field Rendering’.
>
> ## To weakness (5)
>
> >What are the PSNRs reported in Table 1? Are they PSNR after noise removal the same way explored in the baseline, if so the coverage % should also be reported alongside this metric.
>
>  Yes exactly, by default we use 70% coverage, which demonstrates the effectiveness of both methods clearly in the given dataset. We will enclose this configuration in the paper in the revised version.

---

> ### Author Response · Authors · 2024-11-29
>
> ## To weakness (6)
>
> > A few results on NGP-style NeRFs which are the main NeRF models used, and most results on MLP NeRFs.
>
> We clarify this in appendix A.5 and A.7. Essentially, our method is orthogonal to hash encoding, where the input will be mixed with empty space and the model simply renders nothing. However, using a significantly larger hash table is against the objective of NGP-style NeRF — to use less parameters.
>
> Also, Gaussian-splatting based methods provide better rendering quality and speed when there are many training views.
>
> We have to show a few results on NGP-style NeRFs due to the following reasons. First, there was a Hash collision issue caused by the disadvantage of hash encoding. Second, we observed insufficient performance and efficiency of using NGP-style method.
>
> In the revision we would add more results on unbounded cases with Gaussian splatting methods, part of recent result is updated below:
>
> | Object    | Metric | 2DGS 16v | 2DGS 64v | 2DGS 128v | 3DGS 16v | 3DGS 64v | 3DGS 128v |
> |-----------|--------|----------|----------|-----------|----------|----------|-----------|
> | Bicycle   | $\sigma_{max}$   | 0.397    | 0.414    | 0.435     | 0.333    | 0.295    | 0.312     |
> |           | $r_{drop}$  | 0.14     | 0.28     | 0.27      | 0.10     | 0.13     | 0.11      |
> | Bonsai    | $\sigma_{max}$   | 0.471    | 0.398    | 0.357     | 0.425    | 0.376    | 0.290     |
> |           | $r_{drop}$   | 0.28     | 0.33     | 0.36      | 0.20     | 0.25     | 0.27      |
> | Counter   | $\sigma_{max}$   | 0.468    | 0.427    | 0.427     | 0.398    | 0.355    | 0.333     |
> |           | $r_{drop}$   | 0.31     | 0.42     | 0.47      | 0.19     | 0.20     | 0.24      |
> | Flowers   | $\sigma_{max}$   | 0.452    | 0.486    | 0.473     | 0.351    | 0.341    | 0.352     |
> |           | $r_{drop}$   | 0.19     | 0.44     | 0.26      | 0.09     | 0.12     | 0.09      |
> | Garden    | $\sigma_{max}$   | 0.410    | 0.395    | 0.387     | 0.344    | 0.312    | 0.338     |
> |           | $r_{drop}$   | 0.15     | 0.20     | 0.22      | 0.10     | 0.13     | 0.11      |
> | Kitchen   | $\sigma_{max}$   | 0.390    | 0.372    | 0.366     | 0.352    | 0.313    | 0.301     |
> |           | $r_{drop}$   | 0.20     | 0.26     | 0.26      | 0.17     | 0.22     | 0.22      |
> | Room      | $\sigma_{max}$   | 0.469    | 0.390    | 0.394     | 0.362    | 0.322    | 0.284     |
> |           | $r_{drop}$   | 0.37     | 0.41     | 0.47      | 0.22     | 0.27     | 0.29      |
> | Stump     | $\sigma_{max}$  | 0.415    | 0.426    | -         | 0.343    | 0.337    | -         |
> |           | $r_{drop}$   | 0.16     | 0.21     | -         | 0.11     | 0.12     | -         |
> | Treehill  | $\sigma_{max}$   | 0.462    | 0.484    | -         | 0.354    | 0.349    | -         |
> |           | $r_{drop}$  | 0.23     | 0.44     | -         | 0.10     | 0.12     | -         |
>
>
>
> | Object    | Metric | 2DGS 16v | 2DGS 64v | 2DGS 128v | 3DGS 16v | 3DGS 64v | 3DGS 128v |
> |-----------|--------|----------|----------|-----------|----------|----------|-----------|
> | Bicycle   | $\rho_s$     | 0.264    | 0.315    | 0.338     | 0.244    | 0.341    | 0.339     |
> |           | $\rho_p$     | 0.323    | 0.320    | 0.368     | 0.258    | 0.329    | 0.342     |
> | Bonsai    | $\rho_s$     | 0.236    | 0.429    | 0.467     | 0.319    | 0.365    | 0.453     |
> |           | $\rho_p$     | 0.209    | 0.369    | 0.428     | 0.296    | 0.349    | 0.423     |
> | Counter   | $\rho_s$     | 0.304    | 0.524    | 0.536     | 0.399    | 0.423    | 0.454     |
> |           | $\rho_p$     | 0.272    | 0.460    | 0.461     | 0.380    | 0.382    | 0.383     |
> | Flowers   | $\rho_s$     | 0.248    | 0.463    | 0.244     | 0.303    | 0.465    | 0.282     |
> |           | $\rho_p$     | 0.140    | 0.360    | 0.145     | 0.190    | 0.349    | 0.177     |
> | Garden    | $\rho_s$     | 0.112    | 0.305    | 0.310     | 0.129    | 0.314    | 0.307     |
> |           | $\rho_p$     | 0.124    | 0.282    | 0.312     | 0.158    | 0.299    | 0.322     |
> | Kitchen   | $\rho_s$     | 0.243    | 0.386    | 0.389     | 0.288    | 0.433    | 0.411     |
> |           | $\rho_p$     | 0.220    | 0.374    | 0.363     | 0.278    | 0.424    | 0.399     |
> | Room      | $\rho_s$     | 0.383    | 0.477    | 0.499     | 0.386    | 0.443    | 0.423     |
> |           | $\rho_p$     | 0.362    | 0.445    | 0.501     | 0.369    | 0.439    | 0.430     |
> | Stump     | $\rho_s$    | 0.07     | 0.219    | -         | 0.07     | 0.235    | -         |
> |           | $\rho_p$     | 0.162    | 0.290    | -         | 0.109    | 0.277    | -         |
> | Treehill  | $\rho_s$    | 0.273    | 0.414    | -         | 0.284    | 0.419    | -         |
> |           | $\rho_p$     | 0.229    | 0.359    | -         | 0.244    | 0.360    | -         |

---

> ### Author Response · Authors · 2024-11-29
>
> ## To weakness  (7)
>
> > The proof sketches in the main paper can be more detailed.
>
> This will be a major part of revision. Especially in section 4, most of them are not strict proof, which should be either rephrased to intuition or empirical observations.

---

> ### Author Response · Authors · 2024-12-02
> **Reminder for Further Review**
>
> Dear Reviewer,
>
> I apologize for the delay in submitting my response. As today marks the final day of the discussion period, I kindly request your feedback on whether my rebuttal addresses your concerns.
>
> While I am unable to update the PDF at this stage, please rest assured that all revisions will be incorporated into the final version of the paper.
>
> Thank you for your time and consideration.
>
> Best regards,
>
> Authors

---

### Author Response · Authors · 2024-11-29
**Summary of Rebuttal**

Dear Reviewers,

We sincerely thank all reviewers for their valuable feedback. We summarize modifications and the new experiments made in our revised paper.

Apologies for the delayed response to the reviews. We aimed to provide detailed clarifications and updated results wherever possible. We hope the extension allows you to share further comments or questions before the response deadline.

**Key Clarifications and New Results**

We acknowledge the reviewers' concerns regarding the structure of the reasoning in Sections 3 and 4\. We will refine the improper use of proofs, replacing them with intuition-driven analyses for improved clarity. Detailed responses to specific comments are provided below.

We will incorporate new results, including visualizations, to better address unbounded and real-world scenarios. These visual demonstrations will complement the quantitative metrics, enhancing overall understanding. Additionally, we will expand our discussion to include new baselines, such as k-plane.

**Summarized Response to Individual Reviewers**

To Reviewer *k2to*

1. PH-Dropout is not limited to specific test views; it works on arbitrary ones. Following prior works, we use randomly selected test views to illustrate the overall trend.
2. Depth or spatial uncertainty is one of the contributors to epistemic uncertainty. This paper focuses on quantifying overall epistemic uncertainty that includes depth uncertainty. Empirically, our method matches the performance of state-of-the-art depth uncertainty estimation approaches like Bayes Rays while offering additional insights into other aspects of epistemic uncertainty.
3. We would like to clarify that we did not claim our method is "not architecture-dependent," nor can we locate such a statement in our manuscript. We noted that some existing methods are limited to specific architectures like NeRF, while our method generalizes more effectively across NeRF and Gaussian splatting. Nonetheless, we will extend our discussion to include the new method mentioned by the reviewer, such as k-plane.

To Reviewer *6LNs*

1. Hash collision is orthogonal to the neural network after hash encoding, and therefore it is beyond the scope of PH-dropout.
2. To avoid hash collisions, a straightforward approach is to use a larger hash table, though this comes at the cost of reduced efficiency in NGP-style methods.
3. We evaluate PH-dropout on a more real world unbounded dataset, and will add them to the revised version.

To Reviewer *Ptfb*

1. PH-dropout differs from the interpretation in Ledda et al. (2023), as we do not view it as a straightforward approximation to MC-dropout. For example, it requires constraining fidelity on test views during dropout to align with the definition of epistemic uncertainty, which is ignored in Ledda et al. (2023)
2. Our insights are embodied in the feasible dropout operations of PH-dropout, offering step-by-step guidance, including configuring the dropout rate—a critical aspect not addressed by Ledda et al.
3. We agree that certain "proofs" in the paper should instead be framed as intuitions based on existing theorems and new observations. We have addressed each related comment and will thoroughly revise Section 4\.

To Reviewer *pmy8*

1. We have addressed all comments regarding the "theorems" in Sections 3 and 4 and will significantly revise these sections for improved clarity.
2. The revised version will include the previously missing introductions of certain metrics and variables.

To Reviewer *xcSR*

1. PH-Dropout operates solely during the testing phase and does not require scaling, as a constant scale factor does not impact relative uncertainty. This allows it to support downstream applications effectively while slightly reducing computational costs.
2. The evaluation follows the default setup from previous works, aiming to highlight overall trends rather than focusing solely on anchor views.
3. The details of "how Fourier features are dropped" and "why single-layer dropout works" are explained and will be included in the revised version.
4. The reasoning section will be thoroughly revised for improved clarity and soundness.

Thanks again for the valuable feedback.

Authors

---

### Meta-Review · Area_Chair_SRqx · 2024-12-20

**Metareview:**

The evaluated paper deals with the uncertainty estimates added to NERFs and Gaussian splatting, based on test-time drop-out in the lines of Bayesian neural networks. Five different reviewers evaluated the contribution and appreciated the general idea, a strong theoretical foundation and the speed of the method. The main issues raised were,

- Novelty (the approach is very close to / incremental wrt an existing method),
- generality of the method,
- the source of the estimated uncertainty,
- technical soundness and rigor of the proofs,
- Writing of the paper, lack of clarity.

During the reviewers/author discussion phase, the ratings of all reviewers converged to a stable "slightly above borderline". However, these increases anticipated the significant changes promised by the authors, not yet done in the revised version. In the reviewers/AC discussion period, the following assessment was reached: while the method is promising and was appreciated, and most of the questions have been answered by the authors, they failed to provide a revision of the paper itself. Anticipating edits promised by authors is tricky in peer review for any conference, but ICLR has a special open process, which allowed the authors to fully revise their paper. No doing so is a disappointment and provides sufficient cues (or a high risk) that these changes will not happen for the camera ready version. The discussion made it clear, that the reviewers judged the paper to not be ready for publication without these changes. This concerns not only the general presentation of the paper and lack of clarity, but in particular the theoretical aspects, including the quite vague "sketches of proofs". The authors had an opportunity, which they unfortunately missed. For these reasons, the AC judges that the paper is not yet ready for publication.

**Additional Comments On Reviewer Discussion:**

The reviewers engaged with the authors, and discussed the paper with the AC.

---

### Decision · Program_Chairs · 2025-01-22

Reject